



# How big is an OMI pixel?

Martin de Graaf[1,3], Holger Sihler[2], Lieuwe G. Tilstra[3], and Piet Stammes[3]

[1]Delft University of Technology, Delft, The Netherlands
[2]Max-Planck-Institute für Chemie, Mainz, Germany
[3]Royal Netherlands Meteorological Institute, De Bilt, The Netherlands

*Correspondence to:* M. de Graaf, graafdem@knmi.nl

**Abstract.**
The Ozone Monitoring Instrument (OMI) is a push-broom imaging spectrometer, observing solar
radiation backscattered by the Earth's atmosphere and surface. The shape of an OMI pixel is not
quadrangular, which is common for scanning instruments, but rather Gaussian-shaped as light from
neighbouring pixels enters the Field of View (FoV). This has consequences for pixel-area dependent
applications, like cloud fraction products, and visualisation.
The shape and sizes of OMI pixels were determined pre-flight by theoretical and experimental
tests, but never verified after launch. In this paper the OMI point spread function (PSF) is char-
acterised using collocated MODerate resolution Imaging Spectroradiometer (MODIS) reflectance
measurements. MODIS measurements have a much higher spatial resolution than OMI measure-
ments and spectrally overlap at 469 nm. The optimal OMI PSF was determined by finding the highest
correlation between MODIS and OMI reflectances for both cloud-free and partially clouded scenes.
Our results show that the semi-official OMPIXCOR product 75FOV corner coordinates accurately
fix the Full Width at Half Maximum (FWHM) of a super-Gaussian PSF, when this pixel shape is as-
sumed. The exponent of the super-Gaussian PSF is dependent on OMI pixel row number, from about
$n = 2$ at nadir to $3.5$ at the swath edges, due to the increase in pixel size. The optimal Gaussian expo-
nent depends on scene changes between overpasses and reduces to about $n = 1$ for partially clouded
scenes before 2008. Then, the time difference between Aqua and Aura was 15 minutes instead of
8 minutes after 2008. Between overpasses, clouds change the scene, reducing the correlation and
changing the shape of the optimal overlap.
**1 Introduction**
The Ozone Monitoring Instrument (OMI) (Levelt et al., 2006) was launched in 2004 on-board the
Aura satellite, with the main objective to monitor trace gases in the Earth atmosphere, especially
ozone. It was built as the successor to the ESA instruments GOME (Burrows et al., 1999) and
SCIAMACHY (Bovensmann et al., 1999), and NASA's TOMS instruments (e.g. Fleig et al., 1986;
Bhartia et al., 2013). GOME and SCIAMACHY were the first space-borne hyperspectral instru-



ments, measuring the complete spectrum from the ultraviolet (UV) to shortwave-infrared (SWIR)
wavelength range with a relatively high spectral resolution (typically 0.2–1.5 nm), from which mul-
tiple trace gases, clouds and aerosol parameters can be retrieved simultaneously. TOMS instruments
have been monitoring the ozone column at a relatively high spatial resolution ($50\times50$ km$^2$) with
daily global coverage since 1978. OMI was designed to combine those functions and measure the
complete spectrum from the UV to the visible wavelength range (up to 500 nm) with a high spatial
resolution and daily global coverage. To this end, the imaging optics were completely redesigned.

Instead of a rotating mirror, in OMI a two-dimensional CCD detector array ($780\times576$ pixels) is

used to map the incoming radiation in the across-track and wavelength dimensions simultaneously. A
swath of about 2600 km in the across-track direction is imaged along one dimension of the detector
array. Spectrally, the radiation is split into a UV and a visible (VIS) channel and imaged along
the wavelength dimension of the detector array, giving a spectral resolution of 0.63 nm for the VIS
channel. The along-track direction is scanned due to the movement of the satellite. In default 'Global'
operation mode, five consecutive CCD images, each with a nominal exposure time of 0.4 s, are
electronically co-added during a two second interval. The sub-satellite point moves about 13 km
during this time interval (Levelt, 2002). The consequence of this design is that the spatial response
function of the OMI footprints is not box-shaped, but has a peak at the centre of the footprint. This
new design, avoiding moving parts, was used in OMI for the first time, and is now being used in
several new upcoming satellite missions.

The telescope Field of View (FoV) is determined by the projection of the OMI spectrograph

slit on the Earth's surface from the point of view of a CCD pixel. This projection is affected by the
imaging optics and is not a sharply bounded function, but consists of a central response function with
extending tails. The FoV has been determined pre-flight by measuring the intensity response to a star
stimulus for all pixels. This is proprietary information, but the results are summarised here. In the
swath (across-track) direction the average peak position for each pixel was determined and fitted to a
linear curve to determine the spatial sampling distance for the three channels. For the VIS channel the
FoV is 115.1°. The point spread function (PSF) in the across-track direction was not determined (or
reported). However, a memo from the OMI Science Support Team from 2005 shows an across-track
pixel size estimation from these measurements, where the sizes have been determined by assuming
no overlap between adjacent pixels and computing the distances between the peak positions when
imaged on the earth. This yields sizes in the across-track direction of 23.5 km at nadir and 126 km
for far off-nadir (56 degrees) pixels.

In the along-track direction the FoV was characterised by tilting the instrument to simulate the

movement in the flight direction. The measurements were fitted to a normal Gaussian curve with
variable width for different across-track angles and wavelengths. This width is reported as the Full
Width at Half Maximum (FWHM) in degrees, which is about 0.95 at nadir and 1.60 at 56 degrees for
the VIS channel. This corresponds to a nadir pixel size in the along-track direction of about 15 km



and a far off-nadir pixel size of about 42 km, when the Gaussian is convolved with the satellite
motion during 2 s.
The instantaneous FoV (iFoV) of the OMI instrument is influenced by a polarisation scrambler,
that transforms the incoming radiation from one polarisation state into a continuum of polarisation
states (as opposed to unpolarised light). The incoming beam is split into four beams of equal inten-
sity, scrambled, and projected onto the CCD. Since the projections of the four beams are slightly
shifted with respect to each other, the polarisation state of the incoming radiation still slightly deter-
mines the intensity distribution of the four beams and therefore the iFoV in the flight direction. The
only property which is not dependent on the polarisation state of the incoming radiation is the centre
of weight of the four beams. This corresponds to the centre of the ground pixels, which is therefore
the only geolocation coordinate that can be determined unambiguously (van den Oord, 2006).
Therefore, centre coordinates are provided in the Level 1b data product, but corner coordinates are
not. However, for mapping purposes, ground pixel area computations (e.g. for emission estimates per
unit area) and collocation, an OMI corner coordinate product was developed, called OMPIXCOR,
which is provided online via the OMI data portal (Kurosu and Celarier, 2010). Two sets of quadran-
gular corner coordinates are provided. One set contains *tiled* pixel coordinates, which are essentially
the midpoints between adjacent centre coordinates, mainly useful for visualisation purposes, as no
overlap between pixels is imposed. The other set contains so-called 75FOV pixel coordinates, which,
according to Kurosu and Celarier (2010), correspond to 75% of the energy in the along-track FoV.
The authors assumed a 1° FWHM for the iFoV to fix a Gaussian distribution and convolved it with
the satellite movement. The area under a Gaussian curve corresponds to about 76% at FWHM for a
normal distribution (exponent of 2), however, the authors claim to have used a super-Gaussian with
exponent of 4 for this. In this case the energy contained within the FWHM has increased to about
89%. When this iFoV is convolved with the satellite motion, the energy within the FWHM will have
increased even more. The 75FOV pixels generally overlap in the along-track direction, since radi-
ation emanating from adjacent swaths enter the FoV. The coordinates in the across-track direction,
however, are still the half-way points between adjacent pixels.
The application of quadrangular pixel shapes for OMI can become problematic when pixel values
are aggregated onto a regular grid (e.g. Level 3 products that are reported on a regular lat-lon grid).
If pixels overlap, which might occur when several orbits are averaged or in case of 75FOV pixels,
extreme values may be smoothed and reduced due to averaging. A more realistic distribution that
preserves mean values can be reconstructed using a parabolic spline surface on the quadrangular
grid, resulting in a much better visualisation (Kuhlmann et al., 2014). In cases where values from
OMI are compared with that of another instrument, especially with a higher spatial resolution, the
approximate true shape of an OMI pixel is desired. For example, we intend to combine spectral mea-
surements from OMI and MODIS to determine the aerosol direct effect over clouds (de Graaf et al.,



2012). To this end, an optimal characterisation of the PSF of the OMI footprint is desired, to optimise
the accuracy of the retrieval.
In this paper, the OMI PSF for the VIS channel is investigated by testing various predefined
shapes and sizes under various circumstances and determining the maximal correlation between
OMI and MODIS reflectances. In section 2, the consistency between overlapping OMI and MODIS
reflectances is investigated. A cloud-free scene from 2008 is used to study the PSF under the most
optimal circumstances. In chapter 3, a two dimensional super-Gaussian function with a varying expo-
nent is introduced, which can change shape from a near-quadrangular to a sharp-peaked distribution.
Furthermore, the sizes in both along and across-track directions can be varied. This function is used
to define various PSFs, which are investigated for various scenes. The change in PSF is further inves-
tigated by looking into the effect of scene and geometry changes during the (varying) overpass times
of OMI and MODIS. The conclusions from this study are reported in section 4. The geolocations of
the pixels in the UV channels are slightly different from those in the VIS channel. However, the PSF
cannot be determined in the same way for the UV, since MODIS measurements do not overlap with
these channels spectrally.

## 2   Data

Aura flies in formation with Aqua in the Afternoon constellation (A-train). Aqua was launched in
2002, to lead Aura in the A-train by about 15 minutes. The time difference between the instruments
within the A-train is controlled by keeping the various satellites within control-boxes, which are
defined as the maximum distances to which the satellites are allowed to drift before correcting ma-
noeuvres are executed. Therefore, the time difference between OMI and MODIS is variable by up
to a few minutes. A major orbital manoeuvre in 2008 of Aqua decreased the distance between the
Aura and Aqua control boxes to about 8 minutes.
To investigate the correlation between OMI and MODIS observed reflectances, several scenes
were selected. One reference scene will be discussed here in detail. It was an almost cloud-free
scene over the Sahara desert on 4 November 2008, around 14:00 UTC (start of the first MODIS
granule). At this point in time, the time difference between OMI and MODIS was reduced to 8
minutes and around $20 - 30$ seconds, depending on the pixel row. The differences between the pixel
times arise from the fact that MODIS has a scanning mirror, while OMI has no scanning optics, but
exposes the CCD to different scenes while moving in the flight direction. The scene is visualised
in Figure 1, using MODIS channels 2, 1, and 3 to create an RGB picture at 1 km$^2$ resolution. The
MODIS granules are outlined in yellow, while the considered OMI scene is outlined in red. From
June 2007 onward, OMI suffered from a degradation of the observed signal in an increasing number
of rows, called the row anomaly (OMI row anomaly team, 2012). In November 2008 the anomaly
was limited to only rows 53 and 54 for scenes near the equator. These rows were disregarded in the





comparison. In order to stay within the MODIS swath the OMI swath was further reduced to rows 2
to 57. A total of 7,335 OMI pixels are left in the scene.
To compare reflectances from OMI and MODIS, the reflectance measured by OMI is convolved
with the MODIS spectral response function. MODIS channel 3 at 469 nm overlaps with the OMI
VIS channel (350 – 500 nm). This is illustrated in Figure 2, where two OMI reflectance spectra from
the VIS channel are plotted, together with the normalised MODIS response function of channel 3
(red curve). The reflectance spectra correspond to the darkest and brightest pixels (at 469 nm) in
Figure 1, indicated by the green boxes. The darkest pixel is a vegetated area with an OMI reflectance
of 0.0967 and the brightest pixel is a cloud covered scene with an OMI reflectance of 0.5075, both
at 469 nm.
All the 7,335 OMI pixels in the scene in Figure 1 were compared to collocated MODIS pixels, see
the left panel of Figure 3. Here, all the MODIS pixels that fall (partly) within an OMI quadrangular
pixel are averaged with equal weight, which is the easiest and quickest averaging strategy. The
MODIS reflectances show a Pearson's correlation coefficient $r$ of 0.997 with the OMI reflectances,
and a standard deviation (SD) of 0.00433. The MODIS reflectances are somewhat lower than the
OMI reflectances; a linear fit through the points shows a slope of 0.954 and an offset of 0.0010.
**3  OMI point spread function**
The true PSF of an OMI pixel is expected to resemble a flat-top Gaussian shape. To investigate the
OMI PSF, the response at 469 nm is compared to the MODIS channel 3 signals, weighted using
different super-Gaussian functions in two dimensions, and checking the change in the correlation
and SD between the OMI and MODIS reflectances. A 2D super-Gaussian distribution is defined by
$$g(x,y) = \exp\left(-(\frac{x}{w_x})^n - (\frac{y}{w_y})^n\right),$$    (1)
where $x$ and $y$ are the along and across-track directions, and $w_{x,y}$ are the weights in either direction,
defined by
$$w_{x,y} = \frac{\text{FWHM}_{x,y}}{2(\log 2)^{1/n}}.$$    (2)
$\text{FWHM}_{x,y}$ are the full widths at half maximum in the along and across-track directions, respectively,
defined in this paper by the 75FOV pixel corner coordinates. The size of the PSF can be varied
to include more or fewer MODIS pixels from neighbouring pixels in the along and across-track
directions by varying $w_{x,y}$. All size changes are reported relative to $\text{FWHM}_{x,y}$.
The shape of the PSF is determined by the Gaussian exponent $n$, which defines the 'pointed-
ness' of the distribution, $n = 2$ corresponding to a normal distribution, $n < 2$ resulting in a point-hat
distribution and $n > 2$ resulting in a flat-top distribution, see the illustration in one dimension in
Figure 4. Various PSFs are illustrated in Figure 5. The colours of the square MODIS pixels indi-
cate the relative contribution of that pixel. Figure 5a shows the quadrangular OMI pixel, with all



MODIS pixels within the OMI corner coordinates having equal weight, while all pixels outside the
footprint have zero weight. Figure 5b shows a 2D flat-top super-Gaussian ($n = 8$) shape resembling
the quadrangular shape but with smoother edges, and using the 75FOV corner coordinates to fix the
FWHM. Figure 5c shows a normally or 2D Gaussian ($n = 2$) distribution, while Figure 5d shows
a 2D point-hat super Gaussian ($n = 1$) distribution. Figures 5e and f show the weights for pixels
which are assumed to be twice as wide or long as the 75FOV pixels and using a 2D normal Gaussian
distribution.
The size and shape of the assumed PSF was varied in steps of $0.25n$ and $0.25 \cdot$FWHM for a wide
range of these parameters, and for each configuration the correlation between the OMI and MODIS
reflectances and the SD was determined, using all pixels from the scene in Figure 1. The correlation
change is shown in Figure 6. The blue curve shows the change in correlation for a changing exponent
$n$, and $1 \cdot$FWHM, i.e. the change in PSF shape and fixed 75FOV corner coordinates. In this case, the
highest correlation is obtained when a Gaussian distribution with exponent $n = 2.5$ is used, which
is slightly more flat-topped than a normal distribution. The red lines show the change in correlation
when the shape of the distribution is fixed to a normal distribution ($n = 2$). In that case, the corre-
lation peaks for an across-track width of $0.8 \cdot$FWHM, corresponding to a slightly more narrow pixel
in the across-track direction. In the along-track direction the correlation peaks at $1 \cdot$FWHM. If all
three parameters are allowed to vary at the same time, the maximum correlation is found as before:
$n = 2.5$ and the pixel sizes corresponding to the 75FOV corner coordinates in both directions. This
is shown by the purple curve, which shows the variation along the across-track direction for the op-
timal parameters. Obviously, the maximum in the purple curve is the same as the one for the blue
curve: $r = 0.9974$. This is higher than the correlation when quadrangular pixels are used.
The correlation between the OMI and MODIS reflectances and the SD, when the optimal PSF for
this scene is used, is shown in the right panel of Figure 3. The SD for the optimal PSF is 0.00409.
The change in SD for different shapes and sizes is not shown, because it is consistent with the change
of the reciprocal of the correlation, in the sense that it is minimal when the correlation peaks and can
be equally used to find the optimal PSF in this way.

## 3.1 PSF sensitivity

So, when a super-Gaussian form is assumed, the optimal OMI PSF for the reference scene can be
characterised using an exponent $n = 2.5$ and 75FOV corner coordinates for the Gaussian FWHM.
However, the correlation between OMI and MODIS reflectances is not a constant. A number of
scenes were investigated to show the change in correlation between OMI and MODIS reflectances
in time and space. They are treated below and illustrated in Figures 7 – 9.
First, another cloud-free scene was found over the Middle East on 7 October 2008, starting on
10:20 UTC, see Figure 7. The time difference between OMI and MODIS is about 8 minutes and 34–
45 s. This scene is entirely cloud-free over land, and the reflectance ranges from 0.12 over the ocean





to 0.41 over the desert. The correlation between the OMI and MODIS reflectances is depicted in the
right panel of Figure 7, which displays the same dependencies as in Figure 6. The highest correlation
($r = 0.9965$) using 75FOV corner coordinates is found for a Gaussian distribution with an exponent
of $n = 3$ (blue line). When the shape is fixed to a normal distribution ($n = 2$), the highest correlation
($r = 0.9964$) is found for pixel sizes that are smaller ($0.8 \cdot$FWHM) in the across-track direction, as
for the reference scene. This is also the absolute maximum and therefore the red across-track curve
coincides with the purple one.

Next, a scene over Australia was selected on 11 October 2008 starting on 04:45 UTC, see Figure 8.
The time difference between OMI and MODIS is also about 8 minutes and 35–43 s. This scene has
a large cloud-free part. Most cloud pixels, indicated by the red rectangles, were not used in the
analysis. The correlation between OMI and MODIS for various shapes and sizes is again displayed
in the right panel. The maximum correlation for this scene was $r = 0.9907$, obtained for a point-
hat Gaussian distribution with exponent $n = 1.75$ and FWHM corner coordinates. Note that the
correlation is significantly lower than for the reference scene.

Lastly, another Sahara cloud-free scene in the beginning of 2008 was selected, shown in Figure 9.
At this time the correcting manoeuvre bringing OMI closer to MODIS had not yet been performed
and the time difference between the instruments is as large as around 14 minutes, up to 16 minutes
and 26 s. The highest correlation is found for a Gaussian distribution with an exponent of $n = 1.5$
(blue line), which is a point-hat super-Gaussian distribution with wide wings. Similarly, when the
shape is fixed to a normal distribution ($n = 2$), the highest correlation is found for pixel sizes that are
wider than the 75FOV corner coordinates, which is different from the reference scene in Figure 1.
The most striking difference, however, is the much lower absolute value of the correlation. The
maximum correlation for this scene is $r = 0.980$, which is 2% lower than for the reference scene,
in December 2008. Even a 4 times wider pixel size in the reference scene yields a much higher
correlation between the OMI and MODIS. Apparently, the time difference between the Aqua and
Aura of 15 minutes makes a comparison between the two instruments much more challenging, even
for almost cloud-free scenes. It is unlikely that the OMI FoV has changed much between January
and December 2008. Furthermore, a cloud-free Sahara scene in 2006 (31 January 2006, around
13:55 UTC, not shown), showed the same lower correlation, peaking for a Gaussian exponent $n = 1$,
which is also a point-hat distribution with wide tails. The maximum correlation for this scene was
$r = 0.971$, which is in the same order as this scene in January 2008.

## 3.2 Scene dependencies

The effect of changing scenes between overpasses can be illustrated by looking at the pixels with the
highest SD between the OMI reflectances and the average collocated MODIS reflectances. Even for
a scene after 2008, when the overpass time difference is reduced to about 8 minutes, the retrieved
TOA reflectance can change significantly during this time in the case of broken clouds. The pixels





with the highest SD for the reference scene were marked blue in the right panel of Figure 3. The
marked points correspond to the blue coloured OMI pixels in Figure 1, which are the areas where
the scene contains broken cloud fields. In the few minutes between Aqua and Aura overpasses these
clouds change shape and position, changing the average reflectance in a pixel when the cloud fraction
is changed.
This is the main reason for the small optimal super-Gaussian exponent for the 2006 Sahara scene
(Figure 9): due to scene changes during the different overpass times, the observed overlap function
deviates from the true PSF, which closely resembles a Gaussian or flat-topped Gaussian. Instead
a more point-hat distribution with wider wings is found. The centre of the pixel becomes more
important, since this point will still have the highest correlation for both instruments. But since the
signal becomes more spread out, the wider wings give a higher correlation than the true PSF.

## 3.3   Viewing angle dependence

The 2008 Australian scene also has the highest correlation for an exponent smaller than 2, but the
presence of clouds only partly explains this. Most of the cloud pixels were removed, but keeping
those pixels in the correlation experiment increased the optimal Gaussian exponent, to 2.5, rather
than decreasing it. The reason for this is that the OMI PSF is dependent on the pixel row, and the
PSF is wider at the swath ends. Most of the cloud pixels are at the swath ends, and removing these
pixels removes the larger exponents. This viewing angle dependence is treated here.
Since the OMI FoV is dependent on the polarisation of the scene, the PSF should also be depen-
dent on the scattering geometry. To demonstrate this, the OMI PSF was determined as a function of
viewing zenith angle (VZA). For all the scenes described above, the optimal super-Gaussian shape
was determined per OMI pixel row, by varying the Gaussian exponent and determining the maxi-
mum correlation between OMI and MODIS pixels for each pixel row. Then the optimal exponents of
all five scenes presented above were averaged and plotted as a function of pixel row. In this analysis,
the 75FOV pixel sizes were used, to reduce the number of variables and because the above analy-
sis showed that the 75FOV corner coordinates are good indicators of the pixel sizes for Gaussian
shapes. The result is shown in Figure 10. The function shows a very erratic behaviour, due to the
rather large steps in Gaussian exponents nodes that were used ($0.25n$), while the change in correla-
tion for a change in Gaussian exponent is very small near the optimum. As a consequence, the pixel
shape has only a limited sensitivity near the optimum, and the retrieved Gaussian exponent is rather
wildly fluctuating. Averaging over the scenes reduces this, but is somewhat arbitrary. In Figure 10 a
boxcar average over 5 neighbouring points is shown as well.
A general trend can be observed from a flat-topped Gaussian shape towards the edge of the swath
with an exponent of about 3.5 to an exponent of around 2 at nadir. Next to the fact that the OMI
FoV is polarisation dependent, the reason for the increasing exponent towards the swath edges is the
pixel size increase towards the swath edges. The pixel sizes are shown for reference. The OMI pixel





sizes increase dramatically towards the edge for the across-track direction. Wide pixels have smooth
edges and a flat interior, while the small pixels around nadir also have smooth edges, but are too
small to display a flat interior. The left and right edges are just 'glued' together. This is expressed by
a Gaussian exponent of 2 or even lower.
This effect is in the across-track direction only, since the pixel size change in the along-track
direction is much smaller. A Gaussian shape which is fixed in the along-track direction and variable
in the across-track direction will probably give an even higher correlation, but this was not attempted.

### 3.4   Geometry differences

The correlation between OMI and MODIS reflectances at 469 nm shows that OMI reflectances
are consistently about 5% larger than the aggregated MODIS reflectances (see Figure 3). These
differences can be governed by changes in viewing and solar conditions between OMI and MODIS.
Since the optics and sub-satellite points differ for both instruments, the viewing angles are slightly
different, even if the satellites roughly follow the same orbit. More importantly, since Aura is always
behind Aqua, the solar zenith angle for OMI is always different from that of MODIS.
To investigate the effect of the differences in scattering geometry on the measured TOA re-
flectance, a cloud-free Rayleigh reflectance was modelled for each OMI pixel in the reference scene
in Figure 1. Each pixel was simulated twice, once using the OMI scattering geometry and once using
an average MODIS scattering geometry. In this way the expected reflectance difference can be de-
termined due to the difference in overpass time, keeping all else the same. To determine the average
MODIS reflectance, the simulated radiances were averaged over the OMI footprint using the optimal
flat-top Gaussian distribution with $n = 2.5$, as was determined for this scene (Figure 6). The average
radiance was then divided by the cosine of the solar zenith angle of the MODIS pixel which is closest
to the centre of the OMI pixel. In this way, the most representative solar zenith angle is used to nor-
malise the radiances. A realistic surface albedo was taken for each pixel, in order to make the model
results comparable to the observations. The surface albedo database used was the TERRA/MODIS
spatially completed snow-free diffuse bihemispherical land surface albedo database (Moody et al.,
2005). The monochromatic calculations were performed at 469 nm, using a standard Rayleigh at-
mosphere (Anderson et al., 1986) reaching to sea level, and an ozone column of 334 DU. The results
are shown in Figure 12.
The reflectance ranges from about $0.085$ to $0.28$, depending on the surface albedo, which is smaller
than the observed reflectances (cf. Figure 3, right panel). This is mainly due to the clouds in the scene
which are not simulated. The simulated OMI reflectances are larger than the simulated MODIS re-
flectances due to different geometries, like the observations. However, the difference for the simu-
lations, with a slope of $0.9965$ and an offset of $-0.001$, is much smaller than for the observations.
Therefore, we conclude that geometry differences between OMI and MODIS introduce differences
of less than 1% and cannot explain the observed slope between OMI and MODIS reflectances. Most



likely, calibration differences are causing the difference between the observed reflectances. The sim-
ulated correlation and SD are also notably better than for the observed scene. As noted before, clouds
have the largest impact on the correlation between the observed reflectances of a scene.

### 3.5   Accuracy of combining OMI and MODIS

The optimal PSF of OMI can now be determined for practical purposes, i.e. mixed scenes with
ocean, land and clouds. This is needed to determine the accuracy that can be expected when OMI
and MODIS measurements are combined to reconstruct the reflectance spectrum for the entire short-
wave spectrum. To determine the accuracy, the correlation between collocated OMI and MODIS
reflectances and the SD was determined by comparing the instruments for the scene shown in Fig-
ure 11. This scene was taken on 13 June 2006, starting on 13:33 UTC when the time difference
between the instruments was about 15 minutes. The scene contains a mixture of land and ocean
scenes, with and without clouds, and also smoke from biomass burning on the African continent.
Only OMI rows 10–50 were processed, which will often be the case to avoid problems with large
pixels or extreme viewing angles. The optimal correlation was found for a Gaussian exponent $n = 1$
and 75FOV corner coordinates (not shown). The low Gaussian exponent can be explained from the
presence of clouds that change the scene between the overpasses, and the exclusion of wide pixels
at the swath edges. The correlation between the OMI and MODIS reflectances using this shape is
shown in the right panel of Figure 11. Obviously, the correlation is a lot lower than for cloud-free
scenes ($r = 0.963$). The SD is 0.0373, which must be taken into account when OMI and MODIS
reflectances are compared or combined. Furthermore, the slope of a linear fit between the OMI and
MODIS reflectance is 0.909, which is smaller than that for cloud-free scenes, which showed about
5% difference. This larger range in reflectances for cloud scenes apparently off-sets the difference
between the instruments even further.

## 4   Conclusions

The correlation between OMI and collocated MODIS reflectances was determined, to inter-compare
the performance of the instruments and to find the PSF of the OMI footprint. MODIS channel 3 at
469 nm overlaps with OMI's visible channel, and the signals can be compared when the reflectance
signal of OMI is convolved with the MODIS spectral response function, and MODIS reflectances
are aggregated over the OMI footprint.
Due to the design of the OMI CCD detector array and the optical path, the footprint of OMI is
not quadrangular and light from neighbouring pixels enters the OMI FoV. The shape and size of the
footprint was determined for a cloud-free scene, to eliminate as much as possible scene changes due
to the different overpass times of Aura and Aqua. Assuming a super-Gaussian shape with variable
exponent and FWHM, the best characterisation of the OMI PSF is found for an exponent $n = 2 - 2.5$



and 75FOV corner coordinates to define the FWHM. When the corner coordinates are fixed, the Gaussian exponent ranges from about 2 at nadir to about 3.5 at the swath edges. This is partly because the OMI PSF is dependent on polarisation, due to the presence of a polarisation scrambler. Therefore, the OMI PSF changes as a function of viewing angle. However, the main reason is the increase in pixel size for off-nadir angles. For very wide pixels the signal flattens at the centre. This effect may become more pronounced when the super-Gaussian exponent in the across-track direction is made independent of the one in the along-track direction.

The OMI-MODIS overlap function is scene dependent. In particular, for larger time differences between the Aqua and Aura overpasses, the optimal overlap function shape is found for smaller Gaussian exponents $n$, still with the FWHM at the 75FOV corner coordinates. When the scene changes between overpasses the signal is spread over a larger area, centred around the centre coordinate. Therefore, a more optimal overlap function is found for a point-hat distribution with wider wings. This is especially true for cloud scenes, which are most frequent. The correlation decreases, and the SD increases, when clouds are in the scene, and this can be used as an indication of the expected accuracy of a comparison between OMI and MODIS reflectances. For a scene with broken clouds over both land and ocean in 2006, an optimal Gaussian exponent of $n = 1$ was found. However, in general, the changes in correlation coefficient are small for small changes of the Gaussian exponent around 2 (much smaller than e.g. changes due to different time differences). Therefore we recommend that the OMI PSF is approximated by a normal Gaussian distribution with exponent $n = 2$ and 75FOV corner coordinates, as a trade-off between the reduction of the exponent because of scene changes (clouds), and the increase of the exponent at the swath edges.

In all of the investigated cases the OMPIXCOR 75FOV corner coordinates adequately fix the size of the pixel.

The use of non-scanning optics like that used in OMI will be continued in new instruments, in particular TropOMI/Sentinel-5P (Veefkind et al., 2012), to be launched in 2016. For TropOMI, a cloud masking feature is anticipated from Suomi-NPP/VIIRS (Schueler et al., 2002). Sentinel-5P will fly in 'loose formation' with Suomi-NPP, with expected overpass time differences of about 5 minutes. The results from this study are relevant for that mission, since such an overpass time difference will significantly change the overlap function between TropOMI and VIIRS, and affect the accuracy of a cloud mask from VIIRS. High resolution VIIRS measurements can be used in the way presented in the present paper to study and characterise the TropOMI PSF and the accuracy of the cloud mask.

*Acknowledgements.* This project was funded by the Netherlands Space Office, project no.: ALW-GO/12-32. Three anonymous referees are thanked for their constructive remarks on the draft manuscript.



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





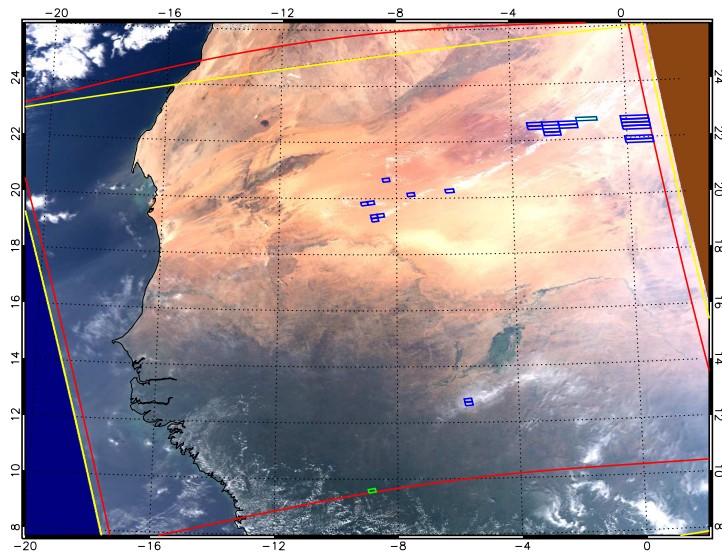

**Figure 1.** MODIS RGB image of the reference scene on 4 November 2008, 14:00 UTC (start of the lower MODIS granule). The yellow lines indicate the MODIS data granules and the red lines the considered OMI swath, which was confined to rows 2 – 57, with the exception of pixels in the row anomaly (see text). The green pixels indicate the darkest (vegetated) and the brightest (cloud covered) areas in the scene. The OMI reflectance spectra of these pixels are shown in Figure 2. The blue OMI pixels correspond to the blue marked points in Figure 3.

Veefkind, J., Aben, I., McMullan, K., Förster, H., de Vries, J., Otter, G., Claas, J., Eskes, H., de Haan, J.,

Kleipool, Q., van Weele, M., Hasekamp, O., Hoogeveen, R., Landgraf, J., Snel, R., Tol, P., Ingmann, P.,

Voors, R., Kruizinga, B., Vink, R., Visser, H., and Levelt, P.: TROPOMI on the ESA Sentinel-5 Precursor:

A GMES mission for global observations of the atmospheric composition for climate, air quality and ozone

layer applications, Remote Sens. Environ., 120, 70–83, 2012.



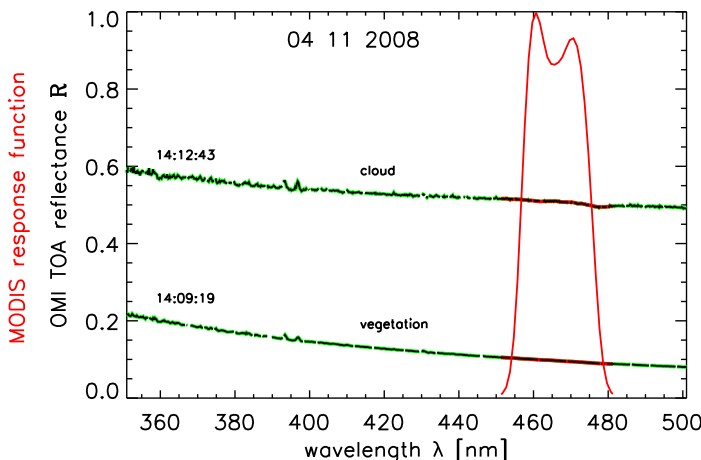

**Figure 2.** OMI top-of-atmosphere reflectance spectra on 4 November 2008, 13:37:24 UTC, and 13:38:02 UTC, of the green pixels in Figure 1 (black/green); and the normalised MODIS response function of channel 3 (red).

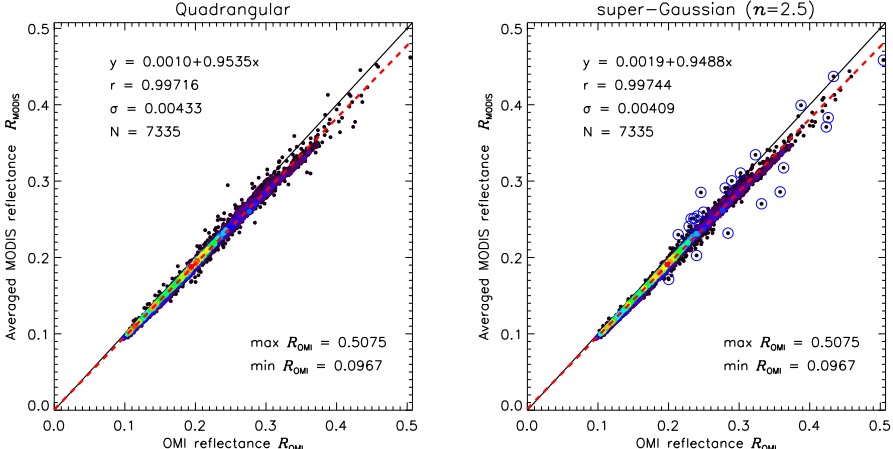

**Figure 3.** Scatter plot of OMI and MODIS collocated reflectances for the scene in Figure 1 using quadrangular OMI pixels (left panel) and optimised super-Gaussian ($n = 2.5$) pixels (right panel). The red dashed line is the linear least squares fit to the measurements, given by the linear function y = $a_0 + a_1 x$ in the plot. $r$ is Pearson's correlation coeffiecient and $\sigma$ the standard deviation. The blue marked points have the largest $\sigma$ and correspond to the blue OMI pixels in Figure 1. N is the number of points and max $R_{OMI}$ and min $R_{OMI}$ the maximum and minimum value in the plot, respectively.





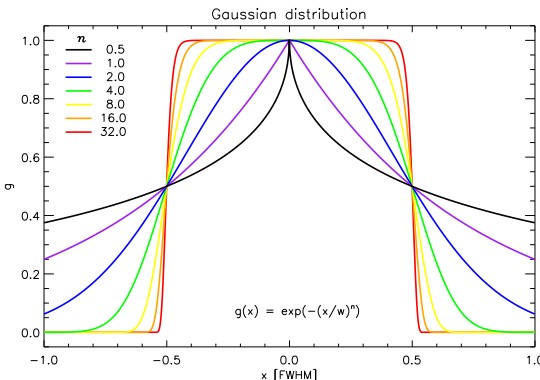

**Figure 4.** One dimensional normalised super-Gaussian distribution functions with varying exponents $n$. The normal distribution ($n = 2$) is plotted in blue.

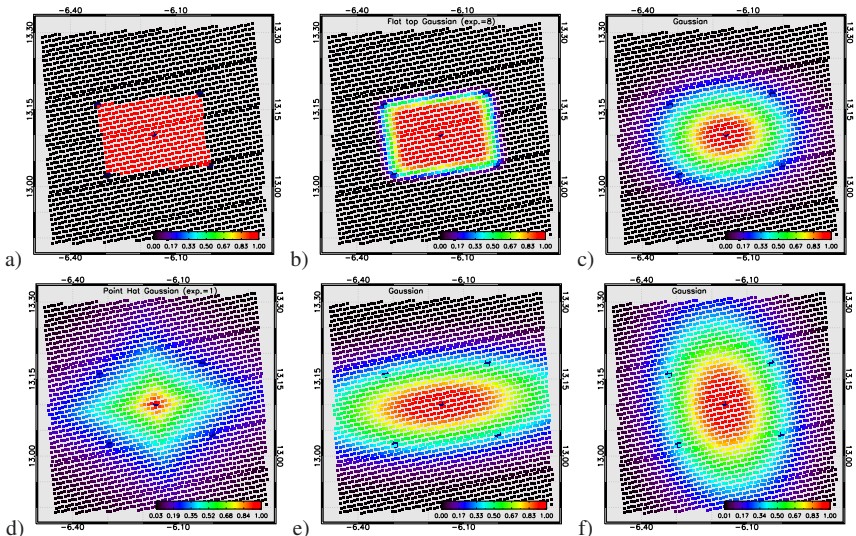

**Figure 5.** OMI 75FOV corner coordinates (dark blue filled circles), with the OMI centre coordinate (dark blue diamond), and collocated MODIS centre coordinates (black and coloured squares). The colours of the squares indicate the weighting of the MODIS pixels as indicated by the colour bar. From top left in reading order: Quadrangular, with all MODIS pixels within the corner coordinates having equal weights, everything else disregarded; a 2D flat top Gaussian with exponent $n = 8$, resembling the rectangle with smoothed edges; a 2D Gaussian or normal distribution; a 2D point-hat Gaussian distribution with exponent $n = 1$; a 2D Gaussian distribution with twice the width in the across-track direction; a 2D Gaussian distribution with twice the width in the along-track direction.





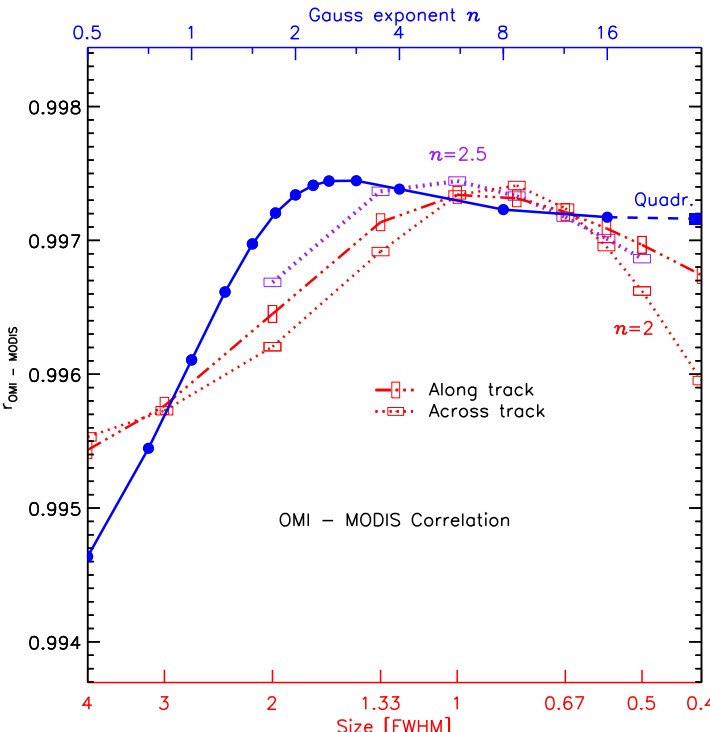

**Figure 6.** Dependence of Pearson's correlation coefficient $r$ between the OMI and MODIS observed reflectance for the scene in Figure 1 as a function of super-Gaussian shape and size. The blue line indicates the relationship as a function of exponent $n$, for fixed 75FOV corner coordinates. The red lines are the relationships for varying pixel sizes when a Gaussian exponent $n = 2$ is chosen. The purple line indicates $r$ as a function of across-track size for the optimal along-track and Gaussian exponent. In this case the optimum was for a Gaussian exponent $n = 2.5$ and 1·FWHM in both directions.





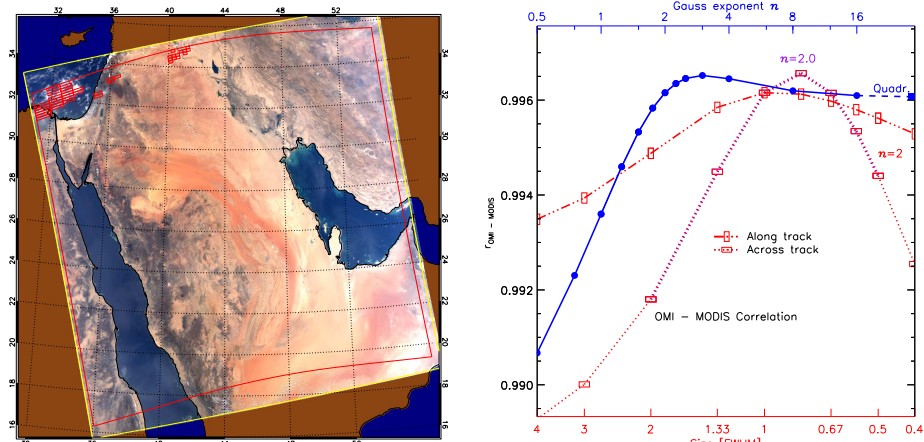

**Figure 7.** Left panel: MODIS RGB scene on on 7 October 2008, 10:20 UTC over the the Middle East. Yellow and red lines as in Figure 1, while the individual red OMI pixels are cloud pixels that were manually discarded. Right panel: Dependence of Pearson's correlation coefficient $r$ between the OMI and MODIS observed reflectance for the scene in the left panel as a function of super-Gaussian shape and size, as in Figure 6. The optimum in this case was found for a Gaussian exponent $n = 3$ and $1 \times 75$FOV corner coordinates in both directions, or $n = 2$ and the across-track size $= 0.8 \times 75$FOV.

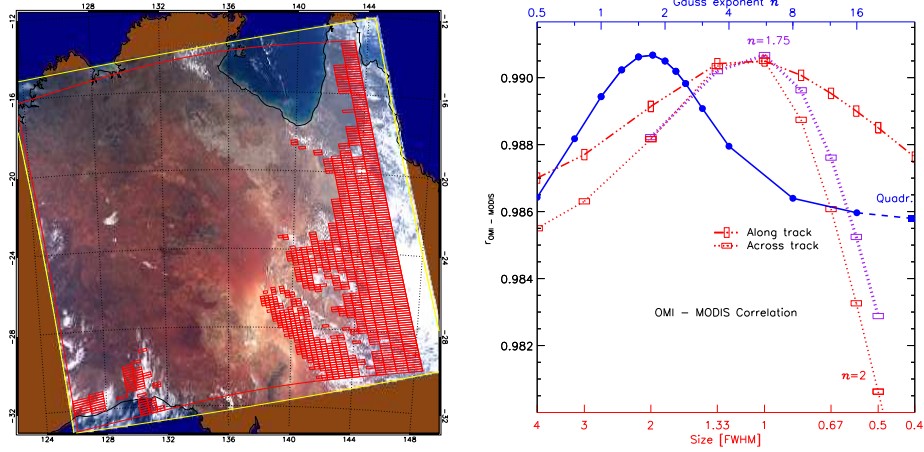

**Figure 8.** Same as Figure 7 on 11 October 2008, 04:45 UTC over Australia. The optimum in this case was found for a Gaussian exponent $n = 1.75$ and $1 \times 75$FOV corner coordinates in both directions.





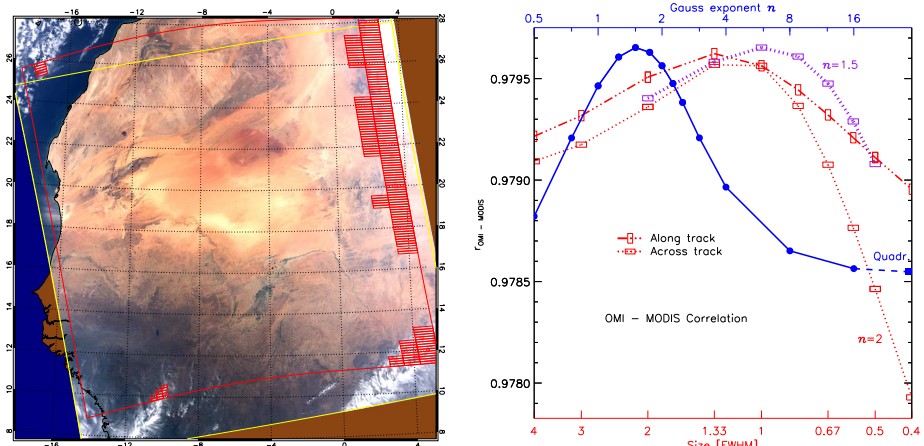

**Figure 9.** Same as Figure 7 on 7 January 2008, 13:45 UTC over the Sahara desert. The optimum in this case was found for a Gaussian exponent $n = 1.5$ and $1 \times 75$FOV corner coordinates in both directions.

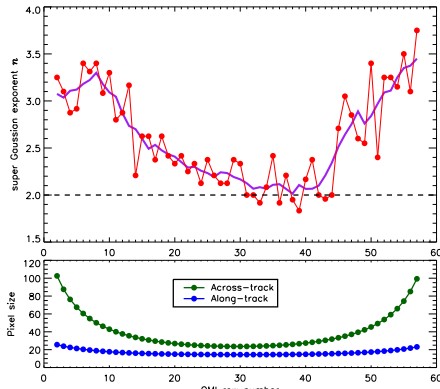

**Figure 10.** Super-Gaussian exponent as a function of OMI pixel row, averaged over all scenes introduced in this paper (red). The FWHM was fixed to the 75FOV pixel sizes, shown in the lower panel, to determine the optimal exponent. The purple curve is the boxcar average of the red curve, using 5 points.





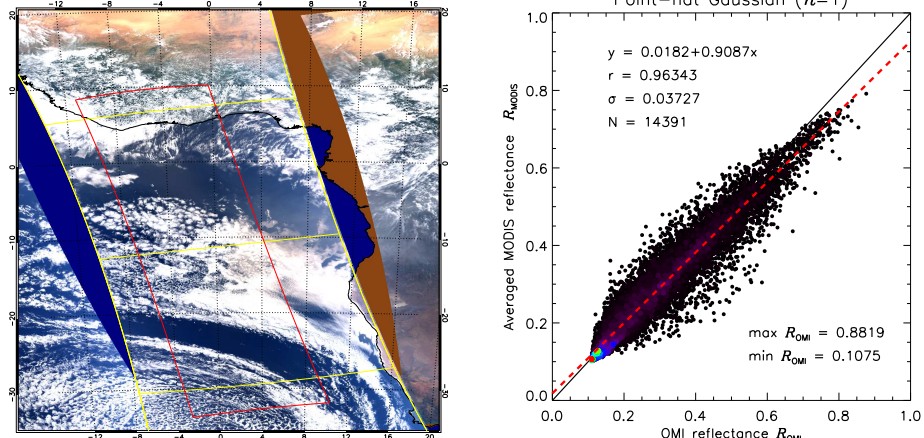

**Figure 11.** MODIS RGB image on 13 August 2006, around 13:33 UTC (lower part of the image). The yellow lines indicate the MODIS data granules and the red lines the considered OMI swath, which was from rows 10 – 50. The optimal correlation between OMI and MODIS for this scene was found for a Gaussian exponent $n = 1$ and 75FOV corner coordinates. The correlation for this pixel shape is shown in the right panel.

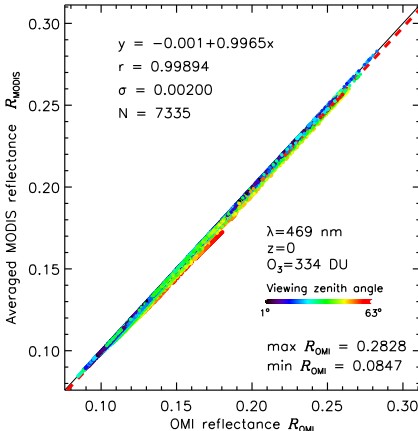

**Figure 12.** Simulated clear-sky reflectances for the reference scene in Figure 1 using OMI scattering geometries ($x$-axis) and MODIS geometries ($y$-axis). The colours indicate the OMI viewing zenith angle of each simulated pixel. The reflectances were simulated at 469 nm, for a standard atmosphere reaching to sea level, and an ozone column of 334 DU. The surface albedo was varied according to a database (see text). The underlying red dashed line shows the linear fit to the simulations.