# Peer review of "How big is an OMI pixel?"

_Atmospheric Measurement Techniques, 2016_

## Referee Comment (RC1) · Anonymous Referee #1 · 21 Apr 2016

Quick Review for **How big is an OMI pixel?** by M. de Graaf, H. Sihler, L.G. Tilstra, and P. Stammes (amt-2016-61)

Graaf et al. present a study on quantifying the spatial size of OMI ground pixels, by matching OMI and co-located MODIS radiances in the visible spectrum. They fit a range of flat-top super-Gaussian spatial functions to match OMI and MODIS under a range of conditions, and compare the results to the OMI OMPIXCOR ground pixel product, the results of which show that the (visible channel) 75FOV OMPIXCOR pixels are a good approximation for the "true" ground pixels as determined by Graaf et al.

The manuscript is clearly organized and well written. It has benefitted greatly from the initial 2015 review and subsequent improvements made by the authors. Thus, very little remains to be criticized. The manuscript is well suited for AMT, and I propose to accept it for publications with a few minor, essentially technical corrections, as outlined below.

**Comments/Corrections**

Page 5, Equation (1):
The 2D super-Gaussian, as stated here, is not the most general form, since both dimensions use the same exponential power $n$. I assume that this is being done to (a) reduce computational requirements for the study, and to (b) use, and compare more easily with, the OMPIXCOR values without having to treat along- and across-track dimensions seperately. Later in the manuscript, the authors make mention of the fact that the two dimensions can be treated independently, but that this hasn't been attempted. I suggest adding a short sentence after Equation (1) to make that point clear right at the place of definition of the super-Gaussian.

Page 5, Equation (2):
Double-check that the weights are correct as written. In particular, whether the power of $1/n$ should not rather be a $1/2$. What prompts me to suggest this is that a Gaussian's Full-Width at Half Maximum (FWHM) and its Half-Width at 1/e (HW1E) are related by

$$\text{HW1e} = \frac{\text{FWHM}}{2\sqrt{\ln(2)}}$$

Page 6, Line 197:
Delete "So, ".

Page 8, Line 246:
"2006 Sahara" should be "2008 Sahara", since the 2006 case is "not shown".

*Page 11, Line 364:*
"changes due to time differences"

*Page 11, Line 370:*
"optics like those of OMI"

*Page 11, Line 377:*
"presented in this paper"

Page 14, Figure 3:
Are the "Quadrangular" OMI pixels from the 75FOV OMPIXCOR product? If so, mention this explicitly since the essentially identical performance of OMPIXCOR and the super-Gaussians are an important result of the paper. If they aren't from OMPIXCOR, add some explanation on the significance of the close performance.

Page 15, Figure 5:

[1] Remove the color bar from each plot and add a larger version outside the individual images. As is, the color bar is too small to read.

[2] Add indications of "(a)", "(b)", …, "(f)" in the figure caption.

Page 19, Figure 12:

[1] Move the color bar outside the figure and make it larger.

[2] As is, this figure conveys very little information, particularly in regards to the color-coded VZA values, since the data points essentially fall on the 1-to-1 line. Here is a suggestion to improve the plot: As X-axis, choose "average" reflectance values R = *(R_MODIS + R_OMI)/2*; these aren't "physical", but they provide a common axis. Against this R, plot the difference in reflectance *dR = R_OMI – R_MODIS*, either absolute or normalized to either *R_MODIS* or *R_OMI*. In that way, the range of the Y axis will become more suitable to the small differences in reflectance, and the color-coding may actually become instructive. *N, y, r,* and *σ* can still be included, as well as the dashed line, though it should be fitted to dR in that case.

---

## Referee Comment (RC2) · Anonymous Referee #3 · 29 Apr 2016

General remarks: This paper tackles an interesting problem: the on-orbit estimation of the size and shape of the point-spread function (PSF) of an OMI field of view (pixel) using collocated data from MODIS, which passes over a ground scene between 8 and 16.5 minutes before OMI does, and measures with a much smaller field-of-view. The idea is a good one.

The authors choose to follow the tradition of approximating the sensitivity function within the FoV by a two-dimensional super-Gaussian function. One should bear in mind that this is a parameterized approximation, and, as such, may fail to deliver a good representation of the actual FoV sensitivity function. It may in fact not be suitable. Rather, it tells one the spatial extent of the bulk of the FoV's integrated sensitivity function and an idea of how "soft" the edges are. Certainly, retrieving values near or less than 1 for the n parameter calls into question either (a) the suitability of this function as a model, or (b) the suitability of the data set to estimate the parameter. The super-

Gaussian function cannot well represent a narrow FoV with a flat top and relatively soft edges, because the extent of its flat top is tied to both its FWHM and its "softness" parameter n. The function chosen, then, may be too highly constrained. The authors have chosen to use a sub-family of functions in which, while the shape is characterised by two width parameters (one along-track, one across-track), the parameter n is constrained to be the same with respect to both these directions. At various points in the discussion, the authors "freeze" the width parameters to be equal to the values they have in the publicly available OMPIXCOR data product, and attempt to optimize the n parameter alone without regard to the sensitivity of their chosen goodness-of-fit statistic, the Pearson correlation coefficient, to the frozen parameters, and how they might move the optimized n value.

In my specific comments below, I note a number of statistical issues that the work has not addressed. Most importantly, no attempt has been made to characterize the uncertainty of the retrieved parameters. This is a serious drawback when comparing results coming from different scenes and different data selection schemes. In this paper, the authors observe that for different scenes, their estimated parameters are quite different. In fact, they ultimately abandon the task of estimating the along-track and cross-track widths in favor of simply accepting the OMPIXCOR FOV75 values, because their data and analysis cannot be used to support a different answer. That is not to say, though that, the OMPIXCOR values are proven by the data. That, then, is the authors' answer to the question posed by the title. The rest is a question of how soft are the sides of the FoVs. The answer to that question is complicated: Figure 10 shows that the answer is scan position dependent (which was not assumed in the calculations up to that point in the paper), and subject to large uncertainty, even when using all available scenes to determine the values of n.

Peer review questions from AMT:

Does the paper address relevant scientific questions within the scope of AMT? – YES
Does the paper present novel concepts, ideas, tools, or data? – YES – Are substantial conclusions reached? –NO– Are the scientific methods and assumptions valid and clearly outlined? NOT AS MUCH AS THEY SHOULD BE Are the results sufficient to support the interpretations and conclusions? – NO – Is the description of experiments and calculations sufficiently complete and precise to allow their reproduction by fellow scientists (traceability of results)? – YES – Do the authors give proper credit to related work and clearly indicate their own new/original contribution? – YES – Does the title clearly reflect the contents of the paper? –YES– Does the abstract provide a concise and complete summary? –YES– Is the overall presentation well structured and clear? –YES– Is the language fluent and precise? –YES– Are mathematical formulae, symbols, abbreviations, and units correctly defined and used? –EXCEPT AS NOTED, YES– Should any parts of the paper (text, formulae, figures, tables) be clarified, reduced, combined, or eliminated? –NO– Are the number and quality of references appropriate? –YES– Is the amount and quality of supplementary material appropriate? –NOT APPLICABLE–

Specific comments:

l. 3: This is awkwardly stated. The "shape" that is not quadrangular is spatial. The projection of a notional field-of-view on the Earth's surface is often thought of as being a quadrangle that we could draw out in latitude-longitude space, for example. What is "Gaussian-shaped" is a section through the sensitivity function of the instrument FoV, as a function of latitude and longitude.

Further, the sentence implies that the "Gaussian-shaped" nature is due to "light from neighbouring pixels enter[ing] the FoV. That is also not an accurate statement: In the absence of measurements made in the adjacent FoVs, a chosen FoV's sensitivity function would remain the same.

The term "pixel" should be either defined or, if it's used synonymously with "FoV," abandoned in favour of the latter, particularly in the Abstract, where it may cause undue confusion.

l. 11: What is meant by "optimal OMI PSF?" The PSF is a physical quantity. I believe "optimal" probably refers to the imposition of a parameterized model for the PSF, and the determination of an optimal set of parameters through the procedure then sketched.

l. 13: Omit "semi-official."

l. 14: I don't think the word "fix" is correct. Do you mean "fits?"

l. 15: I realize that the paper goes on to describe "super-Gaussian" functions, but you have just characterized the same function as "Gaussian."

l. 33: It might help a reader if you mention that these instruments are all in polar, sun-synchronous orbit, so that "global coverage" makes sense.

l. 52: In the previous paragraph, at l. 37, you said the radiation was split into a UV and VIS channel. Here, you refer to three channels. Also, it is not clear what you mean by "spatial sampling distance."

l. 53: What does the number 115.1 deg refer to? Is this actually the field of regard (for the whole swath)? Does it only go from the leftmost to the rightmost FoV centers?

l. 59: The physics behind the shape of the sensitivity function is Fraunhofer diffraction. The classical solution for a circular aperture is an Airy function (with wavelength as a parameter). The use of the relatively simple Gaussian function is as an approximation to the Airy function. The asymmetry of the OMI instrument aperture (along-track and across-track) gives a more complicated geometry, but the diffraction physics is the same. Not saying this suggests that the choice of a Gaussian function is arbitrary.

l. 60: I would suggest removing the word "normal."

l. 64: The satellite motion is not a "function," so it is confusing to say that the Gaussian is convolved with it. It is convolved with a boxcar function whose width is the ~13 km the subsatellite point moves during the 2 second exposure.

l. 81 (and elswhere): Should read "FOV75"

l. 87: See comment at l. 64.

l. 89: "adjacent swaths" is a little confusing, since the word "swath" was used before (e.g., l. 36) to refer to the entire field of regard. (And, of course, there is the confounding use of the word in the context of the data archive.) Perhaps, "successive scans?"

l. 145 and Figure 3: You say sigma is the standard deviation. Standard deviation of what? Is it the RMS deviation of the points from the model line, in the vertical direction? Why would that be preferred to the horizontal direction? That is, your least-squares linear fit to the data is based on the assumption that the OMI reflectances are error-free. Why do you make that assumption? On the right-hand panel of the figure, you highlight the "...points [that] have the largest sigma." What does that mean? Does it mean the S.D. of the reflectances of the MODIS pixels that are collocated to a single OMI FoV? Or the largest deviation of the points' ordinates from the model, along the vertical direction? Or something else?

If there is no discussion of the numerical values characterizing the least-squares linear fit, then why repeat the values in this paragraph, if they are all tabulated in the figure?

l. 152: Why would you assume the same n for the x and y directions in (1)? Is there empirical evidence to support this? If the across-track n turns out to be large, along with w, giving a wider, flatter top in this direction than in the other, the along-track n could still be ∼2, because it is dominated more by diffraction, and less by spacecraft motion. I think this is an important point, and the decision to restrict the functional form in this way deserves solid justification, either in terms of the optical physics or in terms of the empirical data.

l. 167 and Figure 5: The discussion in the text more or less replicates the figure caption. The figure caption would be easier to follow if it used the panel leters (a, b, ..., f), instead of just saying "reading order."

Which OMI row is represented in this figure? The axis orientation changes, FoV by

FoV, as you go across the OMI swath, so the way the MODIS pixels pack into the OMI FoV is different for different FoVs. In Figure 5, you show only 12-15 MODIS pixels along an OMI pixel in the along-track direction. This may be different at wide-of-nadir FoVs, not only because the FoVs are a little larger, but also because the MODIS grid direction cuts through the OMI FoV's x and y directions at a different angle.

l. 176 and Figure 6: If I understand the Figure 6 analyses correctly, you are scaling all the data (for all the different OMI FoVs) to the OMPIXCOR dimensions. If that is so, then your effort is to find optimal wx and wy values that scale *all* OMI FoV positions. I don't know if that is justified. Furthermore, the sizes different OMI FoVs will overlap with different numbers of MODIS pixels, so you may have different uncertainties in r or SD for different FoVs, which would, in turn, bias your optimal wx and wy estimates.

l. 189: Concerning the comparison of the values of r, you do not provide estimates of the uncertainties in your r-values, so can you say that the difference between the r of 0.9974 for the optimized PSF shape and 0.954 for the quadrangle (OMPIXCOR) is significant at some level of confidence?

It appears that your blue curve asymptotes to $\sim$ 0.9972 when n gets large. I am even more skeptical of the implied claim that the difference between 0.9972 and 0.9974 is significant. These are very small differences in a statistic that may be sensitive to sampling artifacts, to the fact that you are using a particular functional form, and the fact that you are constraining the n value to be the same in the along-track and cross-track directions.

Could I ask that you mention, in the caption of Figure 6, that the horizontal scales for all curves (i.e., top and bottom axes) are logarithmic?

l. 192: As I noted before, at l. 145, I do not know how you define SD. I suspect it is the RMS deviation from the best-fit line (perhaps with N-1 in the denominator rather than N, which should not matter much, given your large N). If you are scaling all the FoVs together (see at l. 176), then you should realize different numbers of MODIS pixels

within FoVs at different scan positions. Did you weight the data accordingly (in Figure 3, and its least squares solution)?

If you are using the ordinary least-squares formula (whether weighted or not), you are implicitly assuming that all of your uncertainty is in the MODIS data.

l. 193: I would note, looking at Figure 6, that the maxima in all the curves are pretty broad. That means that your "optimal" values are not very well defined. What would you claim are the uncertainties in the optimized parameters. How are they correlated?

Sec. 3.1: The discussion in this section is problematic. Most of the problems stem from things I have remarked on in the foregoing discussion. The substance of this section is really the difficulty of pinning down values of the optimized parameters. The analysis would be greatly helped by computing uncertainties, including uncertainties due to sampling: How much of the difference you see in the different match-up cases is due to sampling, how much is due to the flatness of the goodness-of-fit functions (r or SD), and how much is due to the way you have chosen to parameterize your PSF model (including the specific choice of a goodness-of-fit metric)?

If you want to claim to have an answer to the question proposed by the paper's title, then indeed the parameter values you obtain should not vary from one case to the next, by more than a certain physically reasonable range.

By the way, in your two Sahara cases, the over-ocean portion of the sample may be quite important: You do not claim to eliminate clouds from either one, and there appear to be large differences between their over-ocean cloud field. The clear scenes over the ocean should contribute very little to the determined slope, since you expect the values there to populate a very limited range in the scatter plots in Figure 3.

You may be correct to attribute the difference between Figure 1 and Figure 9 (and the difficulty, in the latter case, of getting a physically reasonable set of parameters) to the time difference. However, it may simply be that the different scenes have sufficiently

different distributions in their reflectances, and the uncalculated parameter uncertainties are so large that this accounts for a good bit of the difference. I think this may be why you see a notably smaller value of r in the Figure 9 case. In essence, finding a value for n that is near or below 1 challenges the suitability of the super-Gaussian function in (1) to describe the OMI PSF.

Sec. 3.2: To continue my comments from the previous section, you have fixed on the idea that it is the time interval between measurements (8 minutes in Figure 1; 14 to 16.5 min in Figure 9) explains the differences over the broken cloud regions. Certainly, those clouds can evolve significantly on those time scales. But that is not the only possible explanation. For example, MODIS viewing zenith angles (VZA) can be much smaller than those at OMI's off-nadir scenes, and OMI will, in these cases, see a much more homogeneous scene. I would like to suggest that the instantaneous structure of the reflectance field, which is naturally different from one scene to another, may be at least as important as the evolution of the structure from one overpass to the other.

Sec. 3.3: Polarisation may be sufficient to explain the dependence on VZA (i.e., scan position), but it's not necessary. It may be that the diffraction is different at the edges of the far-off-nadir FoVs to those at the edges fo the near-nadir ones.

Figure 10 appears to make my case about the uncertainties. First, by constraining the wx and wy values to the OMPIXCOR (or any) values, you reduce the problem to a single degree of freedom, but if the wx and wy values are not correct, and r is sensitive to them, you have a problem: There is no reasonable physical interpretation of the optimized n value. Furthermore, if you look at the wiggles in the red line in Figure 10, this strongly suggests not erratic behaviour in the instrument or its characterisation vis-a-vis wx and wy, but rather that n is insufficiently well determined, and your scene sampling selects from a parent distribution of n values, and that is different from one scan position to the next.

l. 279: An alternative explanation is that the super-Gaussian may not be an adequate

functional form to describe the FoV PSF. It may also be, as I said before, that an additional problem lies in how you have constrained the wx and wy values.

l. 337: intercompare is one word, no hyphen.

l. 339: This is multiplication, not convolution

l. 344: Set off "as much as possible" with commas.

---

## Author Comment (AC1) · 20 Jun 2016

*Reviewer #1*
*Quick Review for* **How big is an OMI pixel?** *by M. de Graaf, H. Sihler, L.G. Tilstra, and P. Stammes (amt-2016-61)*

*Graaf et al. present a study on quantifying the spatial size of OMI ground pixels, by matching OMI and co-located MODIS radiances in the visible spectrum. They fit a range of flat-top super-Gaussian spatial functions to match OMI and MODIS under a range of conditions, and compare the results to the OMI OMPIXCOR ground pixel product, the results of which show that the (visible channel) 75FOV OMPIXCOR pixels are a good approximation for the true ground pixels as determined by Graaf et al. The manuscript is clearly organized and well written. It has benefitted greatly from the initial 2015 review and subsequent improvements made by the authors. Thus, very*

[Figure]

*little remains to be criticized. The manuscript is well suited for AMT, and I propose to accept it for publications with a few minor, essentially technical corrections, as outlined below.*

The reviewer is thanked for the thorough review and clear assessment. The manuscript has greatly benefitted from this and earlier reviews, and we feel that the manuscript is now in a much better shape, for which we are greatful. The remaining corrections have been performed and addressed below.

**1 Comments/Corrections**

*Page 5, Equation (1):ÂÍThe 2D super-Gaussian, as stated here, is not the most general form, since both dimensions use the same exponential power n. I assume that this is being done to (a) reduce computational requirements for the study, and to (b) use, and compare more easily with, the OMPIXCOR values without having to treat along- and across-track dimensions seperately. Later in the manuscript, the authors make mention of the fact that the two dimensions can be treated independently, but that this hasn't been attempted. I suggest adding a short sentence after Equation (1) to make that point clear right at the place of definition of the super-Gaussian.*

In fact, this was one of the major flaws of the paper, which has been corrected. The super-Gaussian shape has been redefined to use different exponents in along and across-track directions. All the correlations have been recomputed using these new shapes.

*Page 5, Equation (2):ÂÍDouble-check that the weights are correct as written. In particular, whether the power of 1/n should not rather be a 1/2. What prompts me to suggest*

*this is that a Gaussian's Full-Width at Half Maximum (FWHM) and its Half- Width at 1/e (HW1E) are related by*

$$HW1e = \frac{FWHM}{2\mathsf{Sqrt}(ln(2))}$$

The weights in the manuscript are correct. The weights mentioned here are only valid for a normal distribution with n=2. In the new manuscript the FWHM in both directions are now defined separately.

*Page 6, Line 197: Delete So, .* Done.

*Page 8, Line 246:ÂÍ2006 Sahara should be 2008 Sahara, since the 2006 case is not shown.* Changed.

*Page 11, Line 364: changes due to time differences* Changed.

*Page 11, Line 370: optics like those of OMI* Changed.

*Page 11, Line 377: presented in this paper* Changed.

*Page 14, Figure 3:ÂÍAre the Quadrangular OMI pixels from the 75FOV OMPIXCOR product? If so, mention this explicitly since the essentially identical performance of OMPIXCOR and the super-Gaussians are an important result of the paper. If they aren't from OMPIXCOR, add some explanation on the significance of the close performance.* Added.

*Page 15, Figure 5:ÂÍ[1] Remove the color bar from each plot and add a larger version*

[Figure]

*outside the individual images. As is, the color bar is too small to read.Âİ[2] Add indications of (a), (b), ..., (f) in the figure caption.* Done.

*Page 19, Figure 12:Âİ[1] Move the color bar outside the figure and make it larger.Âİ[2] As is, this figure conveys very little information, particularly in regards to the color-coded VZA values, since the data points essentially fall on the 1-to-1 line. Here is a suggestion to improve the plot: As X-axis, choose average reflectance values $R = (R_{MODIS} + R_{OMI})/2$; these aren't physical, but they provide a common axis. Against this R, plot the difference in reflectance $dR = R_{OMI} - R_{MODIS}$, either absolute or normalized to either $R_{MODIS}$ or $R_{OMI}$. In that way, the range of the Y axis will become more suitable to the small differences in reflectance, and the color-coding may actually become instructive. N, y, r, and can still be included, as well as the dashed line, though it should be fitted to dR in that case.*

This is a nice suggestion. We have added it, instead of replacing, even though the new figure conveys no new information compared to the original one, but the original plot clearly shows the behaviour of the simulations compared to the measurements (Figure 3), while the new plot, added as an extra panel, clearly shows the VZA dependence.

---

## Author Comment (AC2) · 20 Jun 2016

Interactive comment on How big is an OMI pixel? by Martin de Graaf et al. Anonymous Referee #3 Received and published: 29 April 2016

General remarks: This paper tackles an interesting problem: the on-orbit estimation of the size and shape of the point-spread function (PSF) of an OMI field of view (pixel) using collocated data from MODIS, which passes over a ground scene between 8 and 16.5 minutes before OMI does, and measures with a much smaller field-of-view. The idea is a good one. The authors choose to follow the tradition of approximating the sensitivity function within the FoV by a two-dimensional super-Gaussian function. One should bear in mind that this is a parameterized approximation, and, as such, may fail to deliver a good representation of the actual FoV sensitivity function. It may in fact

not be suitable. Rather, it tells one the spatial extent of the bulk of the FoVs integrated sensitivity function and an idea of how "soft" the edges are. Certainly, retrieving values near or less than 1 for the n parameter calls into question either (a) the suitability of this function as a model, or (b) the suitability of the data set to estimate the parameter. The super-Gaussian function cannot well represent a narrow FoV with a flat top and relatively soft edges, because the extent of its flat top is tied to both its FWHM and its "softness" parameter n. The function chosen, then, may be too highly constrained. The authors have chosen to use a sub-family of functions in which, while the shape is characterised by two width parameters (one along-track, one across-track), the parameter n is constrained to be the same with respect to both these directions. At various points in the discussion, the authors "freeze" the width parameters to be equal to the values they have in the publicly available OMPIXCOR data product, and attempt to optimize the n parameter alone without regard to the sensitivity of their chosen goodness-of-fit statistic, the Pearson correlation coefficient, to the frozen parameters, and how they might move the optimized n value.

In my specific comments below, I note a number of statistical issues that the work has not addressed. Most importantly, no attempt has been made to characterize the uncertainty of the retrieved parameters. This is a serious drawback when comparing results coming from different scenes and different data selection schemes. In this paper, the authors observe that for different scenes, their estimated parameters are quite different. In fact, they ultimately abandon the task of estimating the along-track and cross-track widths in favor of simply accepting the OMPIXCOR FOV75 values, because their data and analysis cannot be used to support a different answer. That is not to say, though that, the OMPIXCOR values are proven by the data. That, then, is the authors answer to the question posed by the title. The rest is a question of how soft are the sides of the FoVs. The answer to that question is complicated: Figure 10 shows that the answer is scan position dependent (which was not assumed in the calculations up to that point in the paper), and subject to large uncertainty, even when using all available scenes to determine the values of n.
The reviewer is thanked for the very thorough review and many valuable comments. A number of important flaws in the original manuscript have been corrected.

1) The initial choice of parameterization of the FoV sensitivity function, i.e. the 2D super-Gaussian function, was the same in both dimensions, which was a too stringent constraint on the model function to represent the FoV sensitivity. In the paper we suggested to allow different exponents in both directions, and for the current revisions we have indeed implemented this. The new function with an extra free parameter is much more suitable to represent the actual OMI FoV. All the sensitivities have been recomputed, using a super-Gaussian function which is variable in both directions. The new analyses in the revised manuscript therefore give a much more intuitive and clear picture of the OMI FoV.

2) Uncertainties of the correlation parameters and goodness-of-fits have been computed, using (constant) error bars on both OMI and MODIS measurements, while the MODIS uncertainties were weighted with the number of MODIS measurements within a OMI FoV. The goodness-of-fit parameter has been added to plots and discussion.

3) A clearer definition of terms has been used. FoV is used for the projection of the spectrograph slit on the Earth's surface from the point of view of a CCD pixel. A pixel is a detector pixel. The sensitivity function of a FoV is the point spread function (PSF). The overlap function is a 2D super-Gaussian function with the most optimal correlation between OMI and MODIS reflectances for a FoV, dependent on orbital delay and the observed scene.

The PSF of OMI FoV can be approximated relatively consistently using collocated OMI and MODIS reflectances, when carefully selected cloud-free scenes are used with a large enough reflectance range, e.g. desert scenes. The overlap function of
OMI and MODIS reflectances however, is a very complicated function, depending on many parameters, becoming increasingly complex with increasing orbital delay, as we show in the manuscript. We acknowledge the reviewer for the many useful remarks regarding the manuscript, and address each of the comments below.

**1 Peer review questions from AMT**

Does the paper address relevant scientific questions within the scope of AMT? YES Does the paper present novel concepts, ideas, tools, or data? YES Are substantial conclusions reached? NO Are the scientific methods and assumptions valid and clearly outlined? NOT AS MUCH AS THEY SHOULD BE Are the results sufficient to support the interpretations and conclusions? NO Is the description of experiments and calculations sufficiently complete and precise to allow their reproduction by fellow scientists (traceability of results)? YES Do the authors give proper credit to related work and clearly indicate their own new/original contribution? YES Does the title clearly reflect the contents of the paper? YE Does the abstract provide a concise and complete summary? YES Is the overall presentation well structured and clear? YES Is the language fluent and precise? YES Are mathematical formulae, symbols, abbreviations, and units correctly defined and used? EXCEPT AS NOTED. YES Should any parts of the paper (text, formulae, figures, tables) be clarified, re-duced, combined, or eliminated? NO Are the number and quality of references ap- propriate? YES
**2 Specific comments**

I. 3: This is awkwardly stated. The "shape" that is not quadrangular is spatial. The projection of a notional field-of-view on the Earths surface is often thought of as being a quadrangle that we could draw out in latitude-longitude space, for example. What is "Gaussian-shaped" is a section through the sensitivity function of the instrument FoV, as a function of latitude and longitude. Further, the sentence implies that the "Gaussian-shaped" nature is due to "light from neighbouring pixels enter[ing] the FoV. That is also not an accurate statement: In the absence of measurements made in the adjacent FoVs, a chosen FoVs sensitivity function would remain the same. The term "pixel" should be either defined or, if its used synonymously with "FoV," abandoned in favour of the latter, particularly in the Abstract, where it may cause undue confusion.

This indeed may cause confusion. The point to make here was the different sensitivity of the OMI FoV compared to other satellite spectrometers with a scan mirror. The latter instruments often have a FoV sensitivity function that is quadrangular in lat-lon space, with more or less straight, "hard" edges between the corner coordinates designating the limit between very high FoV sensitivity inside these limits and almost negligible sensitivity outside. The OMI FoV sensitivity is markedly different: the corner coordinates give the FWHM of a Gaussian function (or Airy function, but let's assume Gaussian for simplicity) in the along and across-track directions. This gives neither sharply defined sensitivity limits, nor straight lines between corner coordinates in lat-lon space, which is often assumed when talking about pixels or footprints. So, the reviewer
is completely correct here, and the abstract was rewritten according to the suggestions.

*I.* 11: What is meant by "optimal OMI PSF?" The PSF is a physical quantity. I believe "optimal" probably refers to the imposition of a parameterized model for the PSF, and the determination of an optimal set of parameters through the procedure then sketched.

Again, a correct observation. The Point Spread Function (PSF) will be defined as the sensitivity function of the FoV. The 2D super-Gaussian function will be defined as the parameterization of the PSF that is optimized.

**I. 13: Omit "semi-official." done**

**I. 14: I dont think the word "fix" is correct. Do you mean "fits?"**

Actually, it is "fix". The sensitivity of the pixels were determined pre-flight by exposing the pixels to a point source and rotating the instrument. The sensitivity curve found in this way was fitted to a Gaussian curve, of which the FWHM was reported. This FWHM was used to compute the 75FOV corner coordinates. In the present paper, the 75FOV corner coordinates are again used to constrain the super-Gaussian size, by assuming that the FWHM of the Gaussian is the 75FOV width.

However, since this word created more confusion before, the sentence has been rephrased.

*I.* 15: I realize that the paper goes on to describe "super-Gaussian" functions, but you have just characterized the same function as "Gaussian." This has been changed.

I. 33: It might help a reader if you mention that these instruments are all in polar,
*sun-synchronous orbit, so that "global coverage" makes sense.* done.

*I. 52: In the previous paragraph, at I. 37, you said the radiation was split into a UV and VIS channel. Here, you refer to three channels. Also, it is not clear what you mean by "spatial sampling distance."*

The light is actually detected by three channels, two UV and one VIS. This been added. "Spatial sampling distance" is a term copied from the OMI Science Team performance report. It is the distance between the instanteneous FoVs in the across-track direction for individual pixels. This explanation has been added.

*I. 53:* What does the number 115.1 deg refer to? Is this actually the field of regard (for the whole swath)? Does it only go from the leftmost to the rightmost FoV centers? It is actually the field of regard for the whole swath.

*I.* 59: The physics behind the shape of the sensitivity function is Fraunhofer diffraction. The classical solution for a circular aperture is an Airy function (with wavelength as a parameter). The use of the relatively simple Gaussian function is as an approximation to the Airy function. The asymmetry of the OMI instrument aperture (along-track and across-track) gives a more complicated geometry, but the diffraction physics is the same. Not saying this suggests that the choice of a Gaussian function is arbitrary. This is completely correct and this description has been added. The OMI science team report was quoted here, who did not give any more information about the motivation or suitability of the chosen function.

*I. 60: I would suggest removing the word "normal."* done.
*I.* 64: The satellite motion is not a "function," so it is confusing to say that the Gaussian is convolved with it. It is convolved with a boxcar function whose width is the 13 km the subsatellite point moves during the 2 second exposure. Correct, this has been changed.

*I. 81 (and elswhere): Should read "FOV75"* Kurosu and Celarier call this 75FoV pixels, so this was not changed.

*I. 87: See comment at I. 64.* changed.

*I.* 89: "adjacent swaths" is a little confusing, since the word "swath" was used before (e.g., *I.* 36) to refer to the entire field of regard. (And, of course, there is the confounding use of the word in the context of the data archive.) Perhaps, "successive scans?"

Excellent suggestion, this was changed in the manuscript.

I. 145 and Figure 3: You say sigma is the standard deviation. Standard deviation of what? Is it the RMS deviation of the points from the model line, in the vertical direction? Why would that be preferred to the horizontal direction? That is, your least-squares linear fit to the data is based on the assumption that the OMI reflectances are error-free. Why do you make that assumption? On the right-hand panel of the figure, you highlight the "...points [that] have the largest sigma." What does that mean? Does it mean the S.D. of the reflectances of the MODIS pixels that are collocated to a single OMI FoV? Or the largest deviation of the points ordinates from the model, along the vertical direction? Or something else?
It was the RMS deviation of the points from the model line in the vertical direction. This has been changed to the RMS of the points to the model line in both directions.

I. 152: Why would you assume the same n for the x and y directions in (1)? Is there empirical evidence to support this? If the across-track n turns out to be large, along with w, giving a wider, flatter top in this direction than in the other, the along-track n could still be 2, because it is dominated more by diffraction, and less by spacecraft motion. I think this is an important point, and the decision to restrict the functional form in this way deserves solid justification, either in terms of the optical physics or in terms of the empirical data.

This was a major flaw in the previous version and has now been changed.

167 and Figure 5: The discussion in the text more or less replicates the figure caption. The figure caption would be easier to follow if it used the panel leters (a, b, ..., f), instead of just saying "reading order."

The characters have been added.

Which OMI row is represented in this figure? The axis orientation changes, FoV by FoV, as you go across the OMI swath, so the way the MODIS pixels pack into the OMI FoV is different for different FoVs. In Figure 5, you show only 12-15 MODIS pixels along an OMI pixel in the along-track direction. This may be different at wide-of-nadir FoVs, not only because the FoVs are a little larger, but also because the MODIS grid direction cuts through the OMI FoVs x and y directions at a different angle. Correct. Different OMI rows are now shown, to show the changes in orientation and number of MODIS pixels inside one OMI pixel.

I. 176 and Figure 6: If I understand the Figure 6 analyses correctly, you are scaling all
the data (for all the different OMI FoVs) to the OMPIXCOR dimensions. If that is so, then your effort is to find optimal wx and wy values that scale \*all\* OMI FoV positions. I dont know if that is justified. Furthermore, the sizes different OMI FoVs will overlap with different numbers of MODIS pixels, so you may have different uncertainties in r or SD for different FoVs, which would, in turn, bias your optimal wx and wy estimates. Correct. The optimal wx and wy values are relative to the OMPIXCOR values, but the number of MODIS pixels changes. This number has now been used to compute the uncertainties in SD and r.

*I.* 189: Concerning the comparison of the values of *r*, you do not provide estimates of the uncertainties in your *r*-values, so can you say that the difference between the *r* of 0.9974 for the optimized PSF shape and 0.954 for the quadrangle (OMPIXCOR) is significant at some level of confidence?

The goodness-of-fit parameters have been added and reported in the plots.

It appears that your blue curve asymptotes to 0.9972 when n gets large. I am even more skeptical of the implied claim that the difference between 0.9972 and 0.9974 is significant. These are very small differences in a statistic that may be sensitive to sampling artifacts, to the fact that you are using a particular functional form, and the fact that you are constraining the n value to be the same in the along-track and cross-track directions. Could I ask that you mention, in the caption of Figure 6, that the horizontal scales for all curves (i.e., top and bottom axes) are logarithmic? All the sensitivities have been recomputed, now with goodness-of-fit tests and parameters uncertainties, and for varying super-Gaussian parameters in two dimensions. The logarithmic remark has been added.

*I.* 192: As I noted before, at *I.* 145, I do not know how you define SD. I suspect it is the RMS deviation from the best-fit line (perhaps with N-1 in the denominator rather than

AMTD
*N*, which should not matter much, given your large *N*). If you are scaling all the FoVs together (see at l. 176), then you should realize different numbers of MODIS pixels within FoVs at different scan positions. Did you weight the data accordingly (in Figure 3, and its least squares solution)? If you are using the ordinary least-squares formula (whether weighted or not), you are implicitly assuming that all of your uncertainty is in the MODIS data.

This has been changed as noted above.

I. 193: I would note, looking at Figure 6, that the maxima in all the curves are pretty broad. That means that your "optimal" values are not very well defined. What would you claim are the uncertainties in the optimized parameters. How are they correlated? That is correct, the optimized parameters are not very sensitive to changes in Gaussian parameters, see figure 10. This was already mentioned in the manuscript. It is now more clear from the separation of one parameter n into n and m. The uncertainties increase toward larger values of m and n, indicating a better fit for smaller m and n.

Sec. 3.1: The discussion in this section is problematic. Most of the problems stem from things I have remarked on in the foregoing discussion. The substance of this section is really the difficulty of pinning down values of the optimized parameters. The analysis would be greatly helped by computing uncertainties, including uncertainties due to sampling: How much of the difference you see in the different match-up cases is due to sampling, how much is due to the flatness of the goodness-of-fit functions (r or SD), and how much is due to the way you have chosen to parameterize your PSF model (including the specific choice of a goodness-of-fit metric)? If you want to claim to have an answer to the question proposed by the papers title, then indeed the parameter values you obtain should not vary from one case to the next, by more than a certain physically reasonable range. By the way, in your two Sahara cases, the over-ocean portion of the sample may be quite important: You do not claim to
eliminate clouds from either one, and there appear to be large differences between their over-ocean cloud field. The clear scenes over the ocean should contribute very little to the determined slope, since you expect the values there to populate a very limited range in the scatter plots in Figure 3. You may be correct to attribute the difference between Figure 1 and Figure 9 (and the difficulty, in the latter case, of getting a physically reasonable set of parameters) to the time difference. However, it may simply be that the different scenes have sufficiently different distributions in their reflectances, and the uncalculated parameter uncertainties are so large that this accounts for a good bit of the difference. I think this may be why you see a notably smaller value of r in the Figure 9 case. In essence, finding a value for n that is near or below 1 challenges the suitability of the super-Gaussian function in (1) to describe the OMI PSF.

As noted above, all the sensitivities have been recomputed, separating the Gaussian exponents into two dimensions and computing the uncertainties. The goodness-of-fit curve has been added to the figures. This now gives a much clearer picture, and the reviewer is thanked for the suggestions.

As for the Sahara cases: The Sahara desert was chosen because of its bright surface, and as can be seen from Figure 3, the ocean part and the clouds make up the extremes of the reflectance curve, but not the bulk. The bulk of the reflectances are between about 0.1 and 0.3 and they determine most of the fit. Clouds change the SD and correlation, but not the fit so much, because they can drift both into a FoV and out of it, therefore the points with large deviations are distributed about the fit in both directions. The reviewer is correct to suggest that the suitability of the super-Gaussian curve is dubious to characterize the PSF when values of n, m = 1. However, the overlap function doesn't have to resemble the true PSF, especially for large orbital time differences. The change to a point-hat function can be easily explained by the change of the scene reflectance, especially with broken cloud fields.

AMTD
Sec. 3.2: To continue my comments from the previous section, you have fixed on the idea that it is the time interval between measurements (8 minutes in Figure 1; 14 to 16.5 min in Figure 9) explains the differences over the broken cloud regions. Certainly, those clouds can evolve significantly on those time scales. But that is not the only possible explanation. For example, MODIS viewing zenith angles (VZA) can be much smaller than those at OMIs off-nadir scenes, and OMI will, in these cases, see a much more homogeneous scene. I would like to suggest that the instantaneous structure of the reflectance field, which is naturally different from one scene to another, may be at least as important as the evolution of the structure from one overpass to the other.

Sec. 3.3: Polarisation may be sufficient to explain the dependence on VZA (i.e., scan position), but its not necessary. It may be that the diffraction is different at the edges of the far-off-nadir FoVs to those at the edges fo the near-nadir ones.

This is a correct observation and this explanation has been added to the manuscript.

Figure 10 appears to make my case about the uncertainties. First, by constraining the wx and wy values to the OMPIXCOR (or any) values, you reduce the problem to a single degree of freedom, but if the wx and wy values are not correct, and r is sensitive to them, you have a problem: There is no reasonable physical interpretation of the optimized n value. Furthermore, if you look at the wiggles in the red line in Figure 10, this strongly suggests not erratic behaviour in the instrument or its characterisation vis-a-vis wx and wy, but rather that n is insufficiently well determined, and your scene sampling selects from a parent distribution of n values, and that is different from one scan position to the next.

This is correct. n and especially m are not very sensitive to changes near their maximum values. This was already remarked in the previous manuscript, and explained in even more detail in the new manuscript.

AMTD
*I. 279: An alternative explanation is that the super-Gaussian may not be an adequate functional form to describe the FoV PSF. It may also be, as I said before, that an additional problem lies in how you have constrained the wx and wy values.* See the discussions above.

I. 337: intercompare is one word, no hyphen. Corrected.

I. 339: This is multiplication, not convolution. Corrected.

I. 344: Set off "as much as possible" with commas. Done

---

## Author Comment (AC3) · 20 Jun 2016

Manuscript prepared for Atmos. Meas. Tech.
with version 2015/04/24 7.83 Copernicus papers of the LaTeX class copernicus.cls.
Date: 18 June 2016

**How big is an OMI pixel?**

Martin de Graaf[1,3], Holger Sihler[2], Lieuwe G. Tilstra[3], and Piet Stammes[3]

[1]Delft University of Technology, Delft, The Netherlands
[2]Max-Planck-Institute für Chemie, Mainz, Germany
[3]Royal Netherlands Meteorological Institute, De Bilt, The Netherlands

*Correspondence to:* M. de Graaf, graafdem@knmi.nl

**Abstract.**

The Ozone Monitoring Instrument (OMI) is a push-broom imaging spectrometer, observing solar radiation backscattered by the Earth's atmosphere and surface. The incoming radiation is detected using a static imaging CCD detector array with no moving parts, as opposed to most of the previous satellite spectrometers, which used a moving mirror to scan the Earth in the across-track direction.

The sensitivity function of the Field of View (FoV) of detector pixels, projected on the Earth, is defined as the point spread function (PSF). The OMI PSF is not quadrangular, which is common for scanning instruments, but rather super-Gaussian shaped and overlapping with the PSF of neigh- bouring pixels. This has consequences for pixel-area dependent applications, like e.g. cloud fraction products, and visualisation.

The shape and sizes of OMI PSFs were determined pre-flight by theoretical and experimental tests, but never verified after launch. In this paper the OMI PSF is characterised using collocated MODer- ate resolution Imaging Spectroradiometer (MODIS) reflectance measurements. MODIS measure- ments have a much higher spatial resolution than OMI measurements and spectrally overlap at

469 nm. The OMI PSF was verified by finding the highest correlation between MODIS and OMI re- flectances in cloud-free scenes, assuming a 2D super-Gaussian function with varying size and shape to represent the OMI PSF. Our results show that the OMPIXCOR product 75FoV corner coordinates are accurate as the Full Width at Half Maximum (FWHM) of a super-Gaussian PSF model, when this function is assumed. The softness of the function edges, modelled by the super-Gaussian exponents, is different in both directions, and view angle dependent.

The optimal overlap function between OMI and MODIS reflectances is scene dependent, and highly dependent on time differences between overpasses, especially with clouds in the scene. For partially clouded scenes, the optimal overlap function was represented by super-Gaussian exponents around 1 or smaller, which indicates that this function is unsuitable to represent the overlap sensitiv- ity function in these cases. This was especially true for scenes before 2008, when the time differences between Aqua and Aura overpasses was about 15 minutes, instead of 8 minutes after 2008. During the time between overpasses, clouds change the scene reflectance, reducing the correlation and in- fluencing the shape of the optimal overlap function.

**1 Introduction**

The Ozone Monitoring Instrument (OMI) (Levelt et al., 2006) was launched in 2004 on-board the Aura satellite, in a polar, sun-synchronous orbit at approximately 705 km altitude, with a local equatorial crossing-time of 13:45 (ascending node). Its main objective is to monitor trace gases in the Earth atmosphere, especially ozone. It was built as the successor to the ESA instruments GOME (Burrows et al., 1999) and SCIAMACHY (Bovensmann et al., 1999), and NASA's TOMS instruments (e.g. Fleig et al., 1986; Bhartia et al., 2013). GOME and SCIAMACHY were the first spaceborne hyperspectral instruments, measuring the complete spectrum from the ultraviolet (UV) to shortwave-infrared (SWIR) wavelength range with a relatively high spectral resolution (typically 0.2–1.5 nm), from which multiple trace gases, clouds and aerosol parameters can be retrieved simultaneously. TOMS instruments have been monitoring the ozone column at a relatively high spatial resolution ($50 \times 50$ km$^2$) with daily global coverage since 1978. OMI was designed to combine those functions and measure the complete spectrum from the UV to the visible wavelength range (up to 500 nm) with a high spatial resolution and daily global coverage. To this end, the imaging optics were completely redesigned.

Instead of a rotating mirror, in OMI a two-dimensional CCD detector array ($780 \times 576$ pixels) is used to map the incoming radiation in the across-track and wavelength dimensions simultaneously. A swath of about 2600 km in the across-track direction is imaged along one dimension of the detector array. Spectrally, the radiation is split into two UV channels and a visible (VIS) channel and imaged along the wavelength dimension of the detector array. The spectral resolution of the VIS channel is 0.63 nm. The along-track direction is scanned due to the movement of the satellite. In default 'Global' operation mode, five consecutive CCD images, each with a nominal exposure time of 0.4 s, are electronically co-added during a two second interval. The sub-satellite point moves about 13 km during this time interval (Levelt, 2002). The consequence of this design is that the spatial response function of the OMI footprints is not box-shaped, but has a peak at the centre of the footprint. This new design, avoiding moving parts, was used in OMI for the first time, and is now being used in several new upcoming satellite missions.

The telescope Field of View (FoV) is determined by the projection of the OMI spectrograph slit on the Earth's surface from the point of view of a CCD pixel. This projection is affected by Fraunhofer diffraction of the imaging optics, which, for a circular aperture, can be modelled using an Airy function. For a rectangular slit, used in OMI, the solution can be approximated by a Gaussian function in two dimensions. The FoV has been determined pre-flight by measuring the intensity response to a star stimulus for all pixels. The response function was measured by exposing the pixels to a point source and rotating the instrument. The sensitivity curve found in this way was fitted to a Gaussian curve, of which the Full Width at Half Maximum (FWHM) was reported. This is proprietary information, but the results are summarised here. In the swath (across-track) direction the average peak position for each pixel was determined and fitted to a linear curve to determine the spatial sampling distance for the three channels, which gives the instantaneous FoVs in the across-track direction for individual pixels. For the VIS channel the FoV for the entire swath is 115.1°. The point spread function (PSF) in the across-track direction was not determined (or reported). However, a memo from the OMI Science Support Team from 2005 shows an across-track pixel size estimation from these measurements, where the sizes have been determined by assuming no overlap between adjacent pixels and computing the distances between the peak positions when imaged on the earth. This yields sizes in the across-track direction of 23.5 km at nadir and 126 km for far off-nadir (56 degrees) pixels.

In the along-track direction the FoV was characterised by tilting the instrument to simulate the movement in the flight direction. The measurements were fitted to a Gaussian curve with variable width for different across-track angles and wavelengths. This width is reported as the FWHM in degrees, which is about 0.95 at nadir and 1.60 at 56 degrees for the VIS channel. This corresponds to a nadir pixel size in the along-track direction of about 15 km and a far off-nadir pixel size of about 42 km, when the Gaussian is convolved with a boxcar function whose width is the 13 km movement of the subsatellite point during the 2 second exposure.

[revised manuscript text omitted]
 = \frac{\text{FWHM}_x}{2(\log 2)^{1/n}}; \quad w_y = \frac{\text{FWHM}_y}{2(\log 2)^{1/m}}. \qquad (2)$$

FWHM$_{x,y}$ are the full widths at half maximum in the along and across-track directions, respectively,
defined in this paper by the 75FoV pixel corner coordinates. The size of the PSF model can be varied
to include more or fewer MODIS pixels from neighbouring pixels in the along and across-track
directions by varying $w_x$ and $w_y$. All size changes are reported relative to FWHM$_x$ and FWHM$_y$.

The shape of the PSF model is determined by the Gaussian exponents $n$ and $m$, which define
the 'pointedness' of the distribution. In one dimension, $n = 2$ corresponds to a normal distribution,
$n < 2$ results in a point-hat distribution and $n > 2$ results in a flat-top distribution, see the illustration
in Figure 4. Various PSF models are illustrated in Figure 5. The colours of the square MODIS pixels
indicate the relative contribution of that pixel. The different panels show OMI pixels at different
rows, to illustrate the change in orientation and number of MODIS pixels that fall inside an OMI
pixel when the viewing zenith angle changes. Figure 5a shows the quadrangular OMI pixel, with
all MODIS pixels within the OMI corner coordinates having equal weight, while all pixels outside
the footprint have zero weight. Figure 5b shows a 2D flat-top super-Gaussian ($n = m = 8$) shape
using the 75FoV corner coordinates to constrain the FWHM, resembling the quadrangular shape
but with smoother edges. Figure 5c shows a 2D super-Gaussian distribution, with $n = 2, m = 4$,
which represents the optimal representation of the PSF using a super-Gaussian function. Figure 5d
shows a 2D point-hat super-Gaussian ($n = 1, m = 1.5$) distribution, which is the optimal fit of this
function when broken clouds are in the scene. Figures 5e and f show the weights for pixels which are
assumed to be twice as wide or long as the 75FoV pixels and using a 2D super-Gaussian distribution
with $n = 2, m = 4$.

The size and shape of the PSF model was varied by changing $n$ from 0.5 to 16, $m$ from 1 to 16,
and the FWHM from 0.5 to 3 times the 75FoV corner coordinates. For each configuration the corre-
lation between the OMI and MODIS reflectances and the SD were determined, using all pixels from
the scene in Figure 1. The correlation change is shown in Figure 6. The blue dashed-dotted curve
shows the change in correlation for a changing Gaussian exponent and 1·FWHM, i.e. the change
in PSF model shape and 75FoV corner coordinates to constrain the FWHM. In the top panel the
change in correlation coefficient $r$ is shown for a changing Gaussian exponent $n$ using the optimal
Gaussian exponent found for the across-track direction $m = 4$. For this function the optimal Gaus-
sian exponent in the along-track direction is $n = 2$. The blue dotted curve shows the goodness-of-fit
$q$ corresponding to each of the correlation coefficients $r$ (the blue dashed-dotted line). It was deter-
mined using a constant error for OMI measurements, and a constant error for MODIS measurements but weighted by the number of MODIS pixels in each OMI pixel. It shows a reasonably good fit at the optimum $n = 2$.

The red line shows the change in correlation when the along-track width is varied. The shown curve is for the optimal Gaussian parameters, $n = 2, m = 4$, and peaks at 1.0, meaning that the

75FoV corner coordinates are the optimal sizes to constrain the FWHM when a super-Gaussian model is used. The lower panel shows the same dependencies in the across-track direction. The change of $r$ for changing $m$ (the shown dashed-dotted line is for the optimal Gaussian exponent

$n = 2$) and the red curve is the width in the across-track direction for $n = 2, m = 4$. The red curve also peaks at one, again confirming the 75FoV corner coordinates, while $m$ peaks at 4. However, the change for larger $m$ is minimal, meaning that the softness of the edges in the across-track direction make very little difference. Only the goodness-of-fit $q$ decreases significantly for larger $m$, so $m = 4$

can be used as the optimal parameter. These four optimal parameters are also the absolute maximum in the entire parameter space, with $r = 0.998$. This is noticeably higher than the correlation when quadrangular pixels are used.

The correlation between the OMI and MODIS reflectances and the SD, when the optimal PSF

model for this scene is used, is shown in the right panel of Figure 3. The SD for the optimal PSF is

0.0036. The change in SD for different shapes and sizes is not shown, because it is consistent with the change of the reciprocal of the correlation, in the sense that it is minimal when the correlation peaks and can be equally used to find the optimal PSF characterisation in this way.

**227   3.1   PSF sensitivity**

When a super-Gaussian form is assumed, the optimal super-Gaussian model parameters for the refer- ence scene are $n = 2$, $m = 4$ and the 75FoV corner coordinates for the Gaussian FWHM. However, the correlation between OMI and MODIS reflectances is not a constant. A number of scenes were investigated to show the change in correlation between OMI and MODIS reflectances in time and space.

First, another cloud-free scene was found over the Middle East on 7 October 2008, starting on

10:20 UTC, see Figure 7. The time difference between OMI and MODIS is about 8 minutes and

34–45 s. This scene is entirely cloud-free over land, and the reflectance ranges from 0.12 over the ocean to 0.41 over the desert. The correlation between the OMI and MODIS reflectances is depicted in the right panel of Figure 7, which displays the same dependencies as in Figure 6. The highest correlation ($r = 0.9977$) was found for the same super-Gaussian parameters as before, confirming the optimal OMI PSF model. Only the goodness-of-fit was slightly lower than before, indicating a lower correlation between the OMI and MODIS reflectances.

**3.2 Viewing angle dependence**

Next, a scene over Australia was selected on 11 October 2008 starting on 04:45 UTC, see Figure 8. The time difference between OMI and MODIS is about 8 minutes and 35–43 s. This scene has a large cloud-free part, but also a large cloudy part. Most cloud pixels, indicated by the red rectangles, were not used in the analysis. The correlation between OMI and MODIS for various shapes and sizes is again displayed in the right panel. The maximum correlation for this scene was lower than before, $r = 0.9927$, and obtained for a point-hat super-Gaussian distribution with exponents $n = 1.5$ and $m = 2$, and FWHM corner coordinates. The goodness-of-fit is significantly lower than before.

One reason for the lower Gaussian exponents of the 2008 Australian scene in the across-track direction is the removal of the pixels at the end of the swath, which were filtered because of the clouds in those pixels. The OMI PSF is dependent on the pixel row, or viewing angle, with wider PSFs at the swath ends. Since most of the cloud pixels are at the swath ends, removing these pixels removes the larger exponents. The viewing angle dependence of the PSF is treated here.

Since the OMI FoV is dependent on the polarisation of the scene, the PSF should also be dependent on the scattering geometry. Furthermore, the diffraction at the edges of the FoV can be distinctly different for FoVs at nadir compared to those with a large viewing zenith angle (VZA). To investigate this effect, the OMI PSF was characterised using a super-Gaussian function dependent on VZA. For all the scenes described in this paper, the optimal super-Gaussian shape was determined per OMI pixel row, by varying the Gaussian exponent and determining the maximum correlation between OMI and MODIS pixels for each pixel row. Then the optimal exponents were averaged and plotted as a function of pixel row. In this analysis, the 75FoV pixel sizes were used, to reduce the number of variables and because the above analysis showed that the 75FoV corner coordinates are good indicators of the pixel sizes for Gaussian shapes. The result is shown in Figure 9. The super-Gaussian exponents are rather wildly fluctuating, because they have a limited sensitivity near the optimum, especially $m$. Averaging over the scenes reduces this, but is somewhat arbitrary. In Figure 9 a boxcar average over 5 neighbouring points is shown as well.

Still, some change in Gaussian exponents can be observed as a function of VZA. The Gaussian exponent in the across-track direction $m$ changes from around $3-4$ at nadir to about 7 at far off-nadir. Also $n$ is VZA dependent, changing from about $1.5$ at nadir to more than 2 at the swath edges. The reason for the increasing exponents towards the swath edges is the pixel size increase towards the swath edges. The pixel sizes are shown for reference. FoVs at larger VZA are much wider, changing the optimal super-Gaussian that fit the PSF. Furthermore, as observed before, the diffraction at the edges of the FoV will be different at larger viewing angle.

**3.3 Scene dependencies**

The smaller Gaussian exponents for the 2008 Australian scene (Figure 8) are only partly explained by the VZA dependence. The Gaussian exponent $n < 2$ indicates a point-hat super-Gaussian distribution in the along-track direction, which is, as Figure 5e shows, a distribution that is physically unlikely. For this scene, the super-Gaussian function is apparently not a good representation of the PSF of the OMI FoV. The reason for this mismatch are broken cloud fields in the scene, which change the scene reflectance between overpasses of Aqua and Aura. Scene dependencies will be investigated below.

The overpass time between Aqua and Aura changed in 2008, when a correcting manoeuvre brought OMI closer to MODIS. To illustrate the effect, another Sahara cloud-free scene in the beginning of 2008 was selected, when the manoeuvre had not yet been performed, see Figure 10. The time difference between the instruments for this scene is as large as around 14 minutes, up to 16 minutes and 26 s. In this case, the highest correlation is found for a super-Gaussian distribution with exponents $n = 1.5$, $m = 2$, which is again a point-hat super-Gaussian distribution. Similarly, when the shape is fixed to the optimal Gaussian exponents, the highest correlation is found for pixel sizes that are wider than the 75FoV corner coordinates, see the red curves in Figure 10. This is different from the reference scene in Figure 1. The maximum correlation for this scene is $r = 0.982$, which is lower than for the reference scene, in December 2008. The goodness-of-fit $q$ shows much lower values, showing the difficulty with the used PSF model to correlate the OMI and MODIS reflectances. Apparently, the time difference between the Aqua and Aura of 15 minutes makes a comparison between the two instruments much more challenging, even for almost cloud-free scenes. It is unlikely that the OMI FoV has changed much between January and December 2008. Furthermore, a cloud-free Sahara scene in 2006 (31 January 2006, around 13:55 UTC, not shown), showed the same lower correlation, peaking for the same Gaussian exponents.

The effect of changing scenes between overpasses can be illustrated by looking at the pixels with the highest SD between the OMI reflectances and the average collocated MODIS reflectances. Even for a scene after 2008, when the overpass time difference is reduced to about 8 minutes, the retrieved TOA reflectance can change significantly during this time in the case of broken clouds. The pixels with the highest SD for the reference scene were marked blue in the right panel of Figure 3. The marked points correspond to the blue coloured OMI pixels in Figure 1, which are the areas where the scene contains broken cloud fields. In the few minutes between Aqua and Aura overpasses these clouds change shape and position, changing the average reflectance in a pixel when the cloud fraction is changed.

This is the main reason for the small optimal super-Gaussian exponent for the 2008 Sahara scene (Figure 10) and the Australian scene (Figure 8): due to scene changes during the different overpass times, the observed overlap function deviates from the true PSF, which closely resembles a Gaussian or flat-topped Gaussian. Instead a more point-hat distribution with wider wings is found. The centre coordinates have the relative highest correlation, but lower than before, while the correlation becomes smoothed over a larger area, giving the tails of the function a higher correlation than for the true PSF.

**3.4    Accuracy of combining OMI and MODIS**

[revised manuscript text omitted]

reflectances due to different geometries, like the observations. There is a small dependence on VZA, as shown in the right panel of Figure 12, where the relative differences between the OMI and MODIS

reflectances are plotted as a function of either reflectance, to highlight the change for changing VZA

(in colours). However, the difference between the simulated OMI and MODIS reflectances, with a slope of $0.9965$ and an offset of $-0.001$, is much smaller than between the observations. Therefore, we conclude that geometry differences between OMI and MODIS introduce differences of less than

1% and cannot explain the observed slope between OMI and MODIS reflectances. Most likely, calibration differences are causing the difference between the observed reflectances. The simulated correlation and SD are also notably better than for the observed scene. As noted before, clouds have the largest impact on the correlation between the observed reflectances of a scene.

**4 Conclusions**

The correlation between OMI and collocated MODIS reflectances was determined, to intercompare the performance of the instruments and to find the PSF of the OMI footprint. MODIS channel 3 at

469 nm overlaps with OMI's visible channel, and the signals can be compared when the reflectance signal of OMI is multiplied with the MODIS spectral response function, and MODIS reflectances are aggregated over the OMI footprint.

Due to the design of the OMI CCD detector array and the optical path, the footprint of OMI is not quadrangular and light from successive scans enters the OMI FoV. The shape and size of the

PSF of the FoV was determined for a cloud-free scene, to eliminate, as much as possible, scene changes due to the different overpass times of Aura and Aqua. Assuming a super-Gaussian shape with variable exponents and FWHM, the best characterisation of the OMI PSF was found for an exponent $n = 2, m = 4$ and $1 \times 75$FoV corner coordinates to constrain the FWHM.

The OMI PSF changes as a function of viewing angle. When the FWHM are fixed, the Gaussian
exponent ranges from about 1.5 at nadir to more than 2 at the swath edges, while $m$ ranges from about
3-7. This is mainly due to the increase in pixel size for off-nadir angles. Furthermore, the diffraction
at the FoV edges is viewing angle dependent, and the OMI PSF is dependent on polarisation, due to
the presence of a polarisation scrambler in the OMI optical path.

[revised manuscript text omitted]

**Figure 5.** OMI 75FoV corner coordinates (dark blue filled circles), with the OMI centre coordinate (dark blue diamond), and collocated MODIS centre coordinates (black and coloured squares). The colours of the squares indicate the weighting of the MODIS pixels as indicated by the colour bar. a) Quadrangular weighting, with all MODIS pixels within the corner coordinates having equal weights, everything else disregarded; b) a 2D flat-top super-Gaussian with exponents $n = m = 8$, resembling the quadrangular shape with smoothed edges; c) a 2D super-Gaussian distribution with $n = 2$ and $m = 4$; d) a 2D point-hat super-Gaussian distribution with exponents $n = 1, m = 2$; e) a 2D super-Gaussian distribution ($n = 2, m = 4$) with twice the width in the across-track direction; f) a 2D super-Gaussian distribution ($n = 2, m = 4$) with twice the width in the along-track direction. Different OMI row number are shown (see panel captions) to show the change in orientation and number of MODIS pixels for different rows.

[Figure]

**Figure 6.** Pearson's correlation coefficient $r$ for OMI and MODIS collocated reflectances in the scene of Figure 1 as a function of super-Gaussian shape and size of the assumed PSF. The blue line indicates the correlation as a function of exponent $n$ (top panel) and $m$ (lower panel), for fixed 75FoV corner coordinates. The red lines are the relationships for varying pixel sizes when the optimal Gaussian exponents $n = 2, m = 4$ are chosen. Note that the scales are logarithmic on both x-axes.

[Figure]

**Figure 7.** Left panel: MODIS RGB scene on on 7 October 2008, 10:20 UTC over the the Middle East. Yellow and red lines as in Figure 1, while the individual red OMI pixels are cloud pixels that were manually discarded. Right panel: Dependence of Pearson's correlation coefficient $r$ between the OMI and MODIS observed reflectance for the scene in the left panel as a function of super-Gaussian shape and size, as in Figure 6. The optimum in this case was found for Gaussian exponents $n = 2, m = 4$ and $1 \times 75\text{FoV}$ corner coordinates in both directions.

[Figure]

**Figure 8.** Same as Figure 7 on 11 October 2008, 04:45 UTC over Australia. The optimum in this case was found for Gaussian exponents $n = 1.5, m = 2$ and $1 \times 75\text{FoV}$ corner coordinates in both directions. A fit of Gaussian exponents $n = 2, m = 4$ is best for slightly larger pixels ($1.25 \times 75\text{FoV}$, red line).

[Figure]

**Figure 9.** Super-Gaussian exponents $m$ and $n$ as a function of OMI pixel row, averaged over all scenes introduced in this paper. The FWHM was fixed to the 75FoV pixel sizes, shown in the lower panel, to determine the optimal exponent. The fat lines are boxcar averages using 5 points.

[Figure]

**Figure 10.** Same as Figure 7 on 7 January 2008, 13:45 UTC over the Sahara desert. The optimum in this case was found for a Gaussian exponent $n = 1.5, m = 2$ and $1 \times 75$FoV corner coordinates, or $n = 2, m = 4$ and $1.25 \times 75$FoV corner coordinates in both directions.

[Figure]

**Figure 11.** MODIS RGB image on 13 August 2006, around 13:33 UTC (lower part of the image). The yellow lines indicate the MODIS data granules and the red lines the considered OMI swath, which was from rows 10–50. The optimal correlation between OMI and MODIS for this scene was found for Gaussian exponents $n = 1, m = 1.5$ and 75FoV corner coordinates. The correlation for this pixel shape is shown in the right panel.

[Figure]

**Figure 12.** Left panel: Simulated clear-sky reflectances for the reference scene in Figure 1 using OMI scattering geometries ($x$-axis) and MODIS geometries ($y$-axis). The colours indicate the OMI viewing zenith angle of each simulated pixel. The reflectances were simulated at 469 nm, for a standard atmosphere reaching to sea level, and an ozone column of 334 DU. The surface albedo was varied according to a database (see text). The underlying red dashed line shows the linear fit to the simulations. Right panel: same data as in the left panel, but plotted as the relative difference between the OMI and MODIS reflectances.